# DropGNN: Random Dropouts Increase the Expressiveness of Graph Neural Networks

**Pál András Papp**
ETH Zurich
apapp@ethz.ch

**Karolis Martinkus**
ETH Zurich
martinkus@ethz.ch

**Lukas Faber**
ETH Zurich
lfaber@ethz.ch

**Roger Wattenhofer**
ETH Zurich
wattenhofer@ethz.ch

## Abstract

This paper studies Dropout Graph Neural Networks (DropGNNs), a new approach that aims to overcome the limitations of standard GNN frameworks. In DropGNNs, we execute multiple runs of a GNN on the input graph, with some of the nodes randomly and independently dropped in each of these runs. Then, we combine the results of these runs to obtain the final result. We prove that DropGNNs can distinguish various graph neighborhoods that cannot be separated by message passing GNNs. We derive theoretical bounds for the number of runs required to ensure a reliable distribution of dropouts, and we prove several properties regarding the expressive capabilities and limits of DropGNNs. We experimentally validate our theoretical findings on expressiveness. Furthermore, we show that DropGNNs perform competitively on established GNN benchmarks.

## 1 Introduction

Neural networks have been successful in handling various forms of data. Since some of the world's most interesting data is represented by graphs, Graph Neural Networks (GNNs) have achieved state-of-the-art performance in various fields such as quantum chemistry, physics, or social networks [12; 27; 18]. On the other hand, GNNs are also known to have severe limitations and are sometimes unable to recognize even simple graph structures.

In this paper, we present a new approach to increase the expressiveness of GNNs, called Dropout Graph Neural Networks (DropGNNs). Our main idea is to execute not one but *multiple* different runs of the GNN. We then aggregate the results from these different runs into a final result.

In each of these runs, we remove ("drop out") each node in the graph with a small probability $p$. As such, the different runs of an episode will allow us to not only observe the actual extended neighborhood of a node for some number of layers $d$, but rather to observe various slightly perturbed versions of this $d$-hop neighborhood. We emphasize that this notion of dropouts is very different from the popular dropout regularization method; in particular, DropGNNs remove nodes during *both training and testing*, since their goal is to observe a similar distribution of dropout patterns during training and testing.

This dropout technique increases the expressive power of our GNNs dramatically: even when two distinct $d$-hop neighborhoods cannot be distinguished by a standard GNN, their dropout variants (with a few nodes removed) are already separable by GNNs in most cases. Thus by learning to identify the dropout patterns where the two $d$-hop neighborhoods differ, DropGNNs can also distinguish a wide variety of cases that are beyond the theoretical limits of standard GNNs.

**Our contributions.** We begin by showing several example graphs that are not distinguishable in the regular GNN setting but can be easily separated by DropGNNs. We then analyze the theoretical properties of DropGNNs in detail. We first show that executing $\widetilde{O}(\gamma)$ different runs is often already

35th Conference on Neural Information Processing Systems (NeurIPS 2021).

sufficient to ensure that we observe a reasonable distribution of dropouts in a neighborhood of size $\gamma$. We then discuss the theoretical capabilities and limitations of DropGNNs in general, as well as the limits of the dropout approach when combined with specific aggregation methods.

We validate our theoretical findings on established problems that are impossible to solve for standard GNNs. We find that DropGNNs clearly outperform the competition on these datasets. We further show that DropGNNs have a competitive performance on several established graph benchmarks, and they provide particularly impressive results in applications where the graph structure is really a crucial factor.

## 2  Related Work

GNNs apply deep learning to graph-structured data [30]. In GNNs, every node has an embedding that is shared over multiple iterations with its neighbors. This way nodes can gather their neighbors' features. In recent years, many different models have been proposed to realize how the information between nodes is shared [35]. Some approaches take inspiration from convolution [23; 8; 13], others from graph spectra [18; 5], others from attention [33], and others extend previous ideas of established concepts such as skip connections [36].

Principally, GNNs are limited in their expressiveness by the *Weisfeiler-Lehman test* (WL-test) [37], a heuristic to the graph isomorphism problem. The work of [37] proposes a new architecture, *Graph Isomoprhism Networks* (GIN), that is proven to be exactly as powerful as the WL-test. However, even GINs cannot distinguish certain different graphs, namely those that the WL-test cannot distinguish. This finding [11] motivated more expressive GNN architectures. These improvements follow two main paths.

The first approach augments the features of nodes or edges by additional information to make nodes with similar neighborhoods distinguishable. Several kinds of information have been used: inspired from distributed computing are port numbers on edges [28], unique IDs for nodes [20], or random features on nodes [29; 1]. Another idea is to use angles between edges [19] from chemistry (where edges correspond to electron bonds).

However, all of these approaches have some shortcomings. For ports and angles, there are some simple example graphs that still cannot be distinguished with these extensions [11]. Adding IDs or random features helps during training, but the learned models do not generalize: GNNs often tend to overfit to the specific random values in the training set, and as such, they produce weaker results on unseen test graphs that received different random values. In contrast to this, DropGNNs observe a similar distribution of embeddings during training and testing, and hence they also generalize well to test set graphs.

The second approach exploits the fact that running the WL-test on tuples, triples, or generally $k$-tuples keeps increasing its expressiveness. Thus a GNN operating on tuples of nodes has higher expressiveness than a standard GNN [22; 21]. However, the downside of this approach is that even building a second-order graph blows up the graph quadratically. The computational cost quickly becomes a problem that needs to be to addressed, for example with sampling [22]. Furthermore, second-order graph creation is a global operation of the graph that destroys the local semantics induced by the edges. In contrast to this, DropGNN can reason about graphs beyond the WL-test with only a small overhead (through run repetition), while also keeping the local graph structure intact.

Our work is also somewhat similar to the randomized smoothing approach [7], which has also been extended to GNNs recently [4]. This approach also conducts multiple runs on slightly perturbed variants of the data. However, in randomized smoothing, the different embeddings are combined in a smoothing operation (e.g. majority voting), which specifically aims to get rid of the atypical perturbed variants in order to increase robustness. In contrast to this, the main idea of DropGNNs is exactly to find and identify these perturbed special cases which are notably different from the original neighborhood, since these allow us to distinguish graphs that otherwise seem identical.

Finally, we note that removing nodes is a common tool for regularization in deep neural networks, which has also seen use in GNNs [26; 9]. However, as mentioned before, this is a different dropout concept where nodes are only removed during training to reduce the co-dependence of nodes.

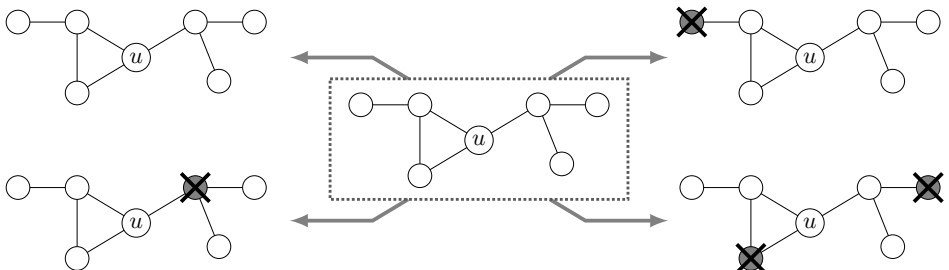

Figure 1: Illustration of 4 possible dropout combinations from an example 2-hop neighborhood around $u$: a 0-dropout, two different 1-dropouts and a 2-dropout.

# 3 DropGNN

## 3.1 About GNNs

Almost all GNN architectures [33; 18; 37; 8; 35; 13; 36] follow the message passing framework [12; 3]. Every node starts with an embedding given by its initial features. One round of message passing has three steps. In the first MESSAGE step, nodes create a message based on their embedding and send this message to all neighbors. Second, nodes AGGREGATE all messages they receive. Third, every node UPDATEs its embedding based on its old embedding and the aggregated messages. One such round corresponds to one GNN layer. Usually, a GNN performs $d$ rounds of message passing for some small constant $d$. Thus, the node's embedding in a GNN reflects its features and the information within its $d$-hop neighborhood. Finally, a READOUT method translates these final embeddings into predictions. Usually, MESSAGE, AGGREGATE, UPDATE and READOUT are functions with learnable parameters, for instance linear layers with activation functions.

This GNN paradigm is closely related to the WL-test for a pair of graphs, which is an iterative color refinement procedure. In rounds $1, ..., d$, each node looks at its own color and the multiset of colors of its direct neighbors, and uses a hash function to select a new color based on this information. As such, if the WL-test cannot distinguish two graphs, then a standard GNN cannot distinguish them either: intuitively, the nodes in these graphs receive the same messages and create the same embedding in each round, and thus they always arrive at the same final result.

## 3.2 Idea and motivation

The main idea of DropGNNs is to execute multiple independent runs of the GNN during both training and testing. In each run, every node of the GNN is removed with probability $p$, independently from all other nodes. If a node $v$ is removed during a run, then $v$ does not send or receive any messages to/from its neighbors and does not affect the remaining nodes in any way. Essentially, the GNN behaves as if $v$ (and its incident edges) were not present in the graph in the specific run, and no embedding is computed for $v$ in this run (see Figure 1 for an illustration).

Over the course of multiple runs, dropouts allow us to not only observe the $d$-hop neighborhood around any node $u$, but also several slightly perturbed variants of this $d$-hop neighborhood. In the different runs, the embedding computed for $u$ might also slightly vary, depending on which node(s) are missing from its $d$-hop neighborhood in a specific run. This increases the expressive power of GNNs significantly: even when two different $d$-hop neighborhoods cannot be distinguished by standard GNNs, the neighborhood variants observed when removing some of the nodes are usually still remarkably different. In Section 3.4, we discuss multiple examples for this improved expressiveness.

Our randomized approach means that in different runs, we will have different nodes dropping out of the GNN. As such, the GNN is only guaranteed to produce the same node embeddings in two runs if we have exactly the same subset of nodes dropping out. Given the $d$-hop neighborhood of a node $u$, we will refer to a specific subset of nodes dropping out as a *dropout combination*, or more concretely as a $k$-dropout in case the subset has size $k$.

In order to analyze the $d$-hop neighborhood of $u$, the reasonable strategy is to use a relatively small dropout probability $p$: this ensures that in each run, only a few nodes are removed (or none at all), and

thus the GNN will operate on a $d$-hop neighborhood that is similar to the original neighborhood of $u$. As a result, 1-dropouts will be frequent, while for a larger $k$, observing a $k$-dropout will be unlikely.

To reduce the effect of randomization on the final outcome, we have to execute multiple independent runs of our GNN; we denote this number of runs by $r$. For a successful application of the dropout idea, we have to select $r$ large enough to ensure that the set of observed dropout combinations is already reasonably close to the actual probability distribution of dropouts. In practice, this will not be feasible for $k$-dropouts with large $k$ that occur very rarely, but we can already ensure for a reasonably small $r$ that e.g. the frequency of each 1-dropout is relatively close to its expected value.

### 3.3  Run aggregation

Recall that standard GNNs first compute a final embedding for each node through $d$ layers, and then they use a READOUT method to transform this into a prediction. In DropGNNs, we also need to introduce an extra phase between these two steps, called *run aggregation*.

In particular, we execute $r$ independent runs of the $d$-layer GNN (with different dropouts), which altogether produces $r$ distinct final embeddings for a node $u$. Hence we also need an extra step to merge these $r$ distinct embeddings into a single final embedding of $u$, which then acts as the input for the READOUT function. This run aggregation method has to transform a multiset of embeddings into a single embedding; furthermore, it has to be a *permutation-invariant* function (similarly to neighborhood aggregation), since the ordering of different runs carries no meaning.

We note that simply applying a popular permutation-invariant function for run aggregation, such as sum or max, is often not expressive enough to extract sufficient information from the distribution of runs. Instead, one natural solution is to first apply a transformation on each node embedding, and only execute sum aggregation afterward. For example, a simple transformation $x \rightarrow \sigma(Wx + b)$, where $\sigma$ denotes a basic non-linearity such as a sigmoid or step function, is already sufficient for almost all of our examples and theoretical results in the paper.

### 3.4  Motivational examples

We discuss several examples to demonstrate how DropGNNs are more expressive than standard GNNs. We only outline the intuitive ideas behind the behavior of the DropGNNs here; however, in Appendix A, we also describe the concrete functions that can separate each pair of graphs.

**Example 1.**  Figure 2a shows a fundamental example of two different graphs that cannot be distinguished by the 1-WL test, consisting of cycles of different length. This example is known to be hard for extended GNNs variants: the two cases cannot even be distinguished if we also use port numbers or angles between the edges [11].

The simplest solution here is to consider a GNN with $d = 2$ layers; this already provides a very different distribution of dropouts in the two graphs. For example, the 8-cycle has 2 distinct 1-dropouts where $u$ retains both of its direct neighbors, but it only has 1 neighbor at distance 2; such a situation is not possible in the 4-cycle at all. Alternatively, the 4-cycle has a 1-dropout case with probability $p \cdot (1-p)^2$ where $u$ has 2 direct neighbors, but no distance 2 neighbors at all; this only happens for a 2-dropout in the 8-cycle, i.e. with a probability of only $p^2 \cdot (1-p)^2$. With appropriate weights, a GNN can learn to recognize these situations, and thus distinguish the two cases.

**Example 2.**  Figure 2b shows another example of two graphs that cannot be separated by a WL test; note that node features simply correspond to the degrees of the nodes. From an algorithmic perspective, it is not hard to distinguish the two graphs from specific 1-dropout cases. Let $u$ and $v$ denote the two gray nodes in the graphs, and consider the process from $u$'s perspective. In both graphs, $u$ can recognize if $v$ is removed in a run since $u$ does not receive a "gray" message in the first round. However, the dropout of $v$ has a different effect in the two graphs later in the process: in the right-hand graph, it means that there is no gray neighbor at a 3-hop distance from $u$, while in the left-hand graph, $u$ will still see a gray node (itself) in a 3-hop distance.

Thus by identifying the 1-dropout of $v$, an algorithm can distinguish the two graphs: if we observe runs where $u$ receives no gray message in the first round, but it receives an (aggregated) gray message

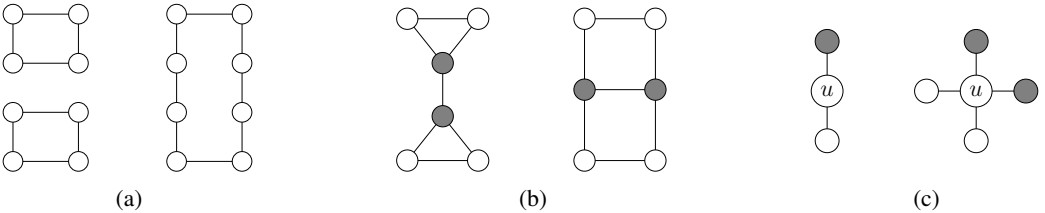

Figure 2: Several example graphs which show that DropGNNs are more expressive than standard GNNs in various cases. Different node colors correspond to different node features.

in the third round, then $u$ has the left-hand neighborhood. This also means that a sufficiently powerful GNN which is equivalent to the 1-WL test can also separate the two cases.

**Example 3.** Note that using a `sum` function for neighborhood aggregation is often considered a superior choice to `mean`, since $u$ cannot separate e.g. the two cases shown in Figure 2c with `mean` aggregation [37]. However, the `mean` aggregation of neighbors also has some advantages over `sum`; most notably, it the computed values do not increase with the degree of the node.

We show that dropouts also increase the expressive power of GNNs with `mean` aggregation, thus possibly making `mean` aggregation a better choice in some applications. In particular, a DropGNN with `mean` aggregation is still able to separate the two cases on Figure 2c.

Assume that the two colors in the figure correspond to feature values of 1 and $-1$, and let $p = \frac{1}{4}$. In the left-hand graph, there is a 1-dropout where $u$ ends up with a single neighbor of value 1; hence mean aggregation yields a value of 1 with probability $\frac{1}{4} \cdot \frac{3}{4} \approx 0.19$ in each run. However, in the right-hand graph, the only way to obtain a mean of 1 is through a 2-dropout or some 3-dropouts; one can calculate that the total probability of these is only 0.06 (see Appendix A). If we first transform all other values to 0 (e.g. with $\sigma(x - 0.5)$, where $\sigma$ is a step function), then run aggregation with `mean` or `sum` can easily separate these cases. Note that if we apply a more complex transformation at run aggregation, then separation is even much easier, since e.g. the mean value of 0.33 can only appear in the right-hand graph.

## 4 Theoretical analysis

### 4.1 Required number of runs

We analyze DropGNNs with respect to the *neighborhood of interest* around a node $u$, denoted by $\Gamma$. That is, we select a specific region around $u$, and we want to ensure that the distribution of dropout combinations in this region is reasonably close to the actual probabilities. This choice of $\Gamma$ then determines the ideal choice of $p$ and $r$ in our DropGNN.

One natural choice is to select $\Gamma$ as the entire $d$-hop neighborhood of $u$, since a GNN will always compute its final values based on this region of the graph. Note that even for this largest possible $\Gamma$, the size of this neighborhood $\gamma := |\Gamma|$ does not necessarily scale with the entire graph. That is, input graphs in practice are often sparse, and we can e.g. assume that their node degrees are upper bounded by a constant; this is indeed realistic in many biological or chemical applications, and also a frequent assumptions in previous works [28]. In this case, having $d = O(1)$ layers implies that $\gamma$ is also essentially a constant, regardless of the size of the graph.

However, we point out that $\Gamma$ can be freely chosen as a neighborhood of any specific size. That is, even if a GNN aggregates information within a distance of $d = 5$ layers, we can still select $\Gamma$ to denote, for example, only the 2-hop neighborhood of $u$. The resulting DropGNN will still compute a final node embedding based on the entire 5-hop neighborhood of $u$; however, our DropGNN will now only ensure that we observe a reasonable distribution of dropout combinations in the 2-hop neighborhood of $u$.

In this sense, the size $\gamma$ is essentially a trade-off hyperparameter: while a smaller $\gamma$ will require a smaller number of runs $r$ until the distribution of dropout combinations stabilizes, a larger $\gamma$ allows us to observe more variations of the region around $u$.

**1-complete dropouts.** From a strictly theoretical perspective, choosing a sufficiently large $r$ always allows us to observe every possible dropout combination. However, since the number of combinations is exponential in $\gamma$, this approach is not viable in practice (see Appendix B for more details).

To reasonably limit the number of necessary runs, we focus on the so-called 1-*complete case*: we want to have enough runs to ensure that at least every 1-dropout is observed a few times. Indeed, if we can observe each variant of $\Gamma$ where a single node is removed, then this might already allow a sophisticated algorithm to reconstruct a range of useful properties of $\Gamma$. Note that in all of our examples, a specific 1-dropout was already sufficient to distinguish the two cases.

For any specific node $v \in \Gamma$, the probability of a 1-dropout for $v$ is $p \cdot (1 - p)^{\gamma}$ in a run (including the probability that $u$ is not dropped out). We apply the $p$ value that maximizes the probability of such a 1-dropout; a simple differentiation shows that this maximum is obtained at $p^* = \frac{1}{1+\gamma}$.

This choice of $p$ also implies that the probability of observing a specific 1-dropout in a run is

$$\frac{1}{1+\gamma} \cdot \left(\frac{\gamma}{1+\gamma}\right)^{\gamma} \geq \frac{1}{1+\gamma} \cdot \frac{1}{e}.$$

Hence if we execute $r \geq e \cdot (\gamma + 1) = \Omega(\gamma)$ runs, then the expected number of times we observe a specific 1-dropout (let us denote this by $\mathbb{E}_1$) is at least $\mathbb{E}_1 \geq r \cdot \frac{1}{e} \cdot \frac{1}{1+\gamma} \geq 1$.

Moreover, one can use a Chernoff bound to show that after $\Omega(\gamma \log \gamma)$ runs, the frequency of each 1-dropout is sharply concentrated around $\mathbb{E}_1$. This also implies that we indeed observe each 1-dropout at least once with high probability.

For a more formal statement, let us consider a constant $\delta \in [0, 1]$ and an error probability $\frac{1}{t} < 1$. Also, given a node $v \in \Gamma$ (or subset $S \subseteq \Gamma$), let $X_v$ (or $X_S$) denote the number of times this 1-dropout ($|S|$-dropout) occurs during our runs.

**Theorem 1** *If $r \geq \Omega\left(\gamma \log \gamma t\right)$, then with a probability of $1 - \frac{1}{t}$, it holds that for each $v \in \Gamma$, we have $X_v \in \left[\, (1 - \delta) \cdot \mathbb{E}_1 \,,\; (1 + \delta) \cdot \mathbb{E}_1 \,\right]$.*

With slightly more runs, we can even ensure that each $k$-dropout for $k \geq 2$ happens less frequently than 1-dropouts. In this case, it already becomes possible to distinguish 1-dropouts from multiple-dropout cases based on their frequency.

**Theorem 2** *If $r \geq \Omega\left(\gamma^2 + \gamma \log \gamma t\right)$, then with a probability of $1 - \frac{1}{t}$ it holds that*

- *for each $v \in \Gamma$, we have $X_v \in \left[\, (1 - \delta) \cdot \mathbb{E}_1 \,,\; (1 + \delta) \cdot \mathbb{E}_1 \,\right]$,*

- *for each $S \subseteq \Gamma$ with $|S| \geq 2$, we have $X_S < (1 - \delta) \cdot \mathbb{E}_1$.*

Since the number of all dropout combinations is in the magnitude of $2^{\gamma}$, proving this bound is slightly more technical. We discuss the proofs of these theorems in Appendix B.

Note that in sparse graphs, where $\gamma$ is essentially a constant, the number of runs described in Theorems 1 and 2 is also essentially a constant; as such, DropGNNs only impose a relatively small (constant factor) overhead in this case.

Finally, note that these theorems only consider the dropout distribution around a specific node $u$. To ensure the same properties for all $n$ nodes in the graph simultaneously, we need to add a further factor of $n$ within the logarithm to the number of necessary runs in Theorems 1 and 2. However, while this is only a logarithmic dependence on $n$, it might still be undesired in practice.

### 4.2 Expressive power of DropGNNs

In Section 3.4, we have seen that DropGNNs often succeed when a WL-test fails. It is natural to wonder about the capabilities and limits of the dropout approach in general; we study this question for multiple neighborhood aggregation methods separately.

We consider neighborhood aggregation with `sum` and `mean` in more detail; the proofs of the corresponding claims are discussed in Appendices C and D, respectively. Appendix D also discusses briefly why `max` aggregation does not combine well with the dropout approach in practice.

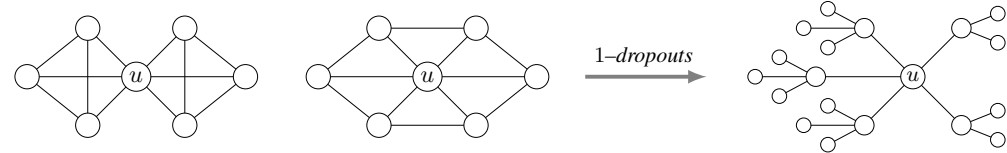

Figure 3: Example of two graphs not separable by 1-dropouts (left side). In both of the graphs, for any of the 1-dropouts, $u$ observes the same tree structure for $d = 2$, shown on the right side.

**Aggregation with `sum`.** Previous work has already shown that `sum` neighborhood aggregation allows for an injective GNN design, which computes a different embedding for any two neighborhoods whenever they are not equivalent for the WL-test [37]. Intuitively speaking, this means that `sum` aggregation has the same expressive power as a general-purpose $d$-hop distributed algorithm in the corresponding model, i.e. without IDs or port numbers. Hence to understand the expressiveness of DropGNNs in this case, one needs to analyze which embeddings can be computed by such a distributed algorithm from a specific (observed) distribution of dropout combinations.

It is already non-trivial to find two distinct neighborhoods that cannot be distinguished in the 1-complete case. However, such an example exists, even if we also consider 2-dropouts. That is, one can construct a pair of $d$-hop neighborhoods that are non-isomorphic, and yet they produce the exact same distribution of 1- and 2-dropout neighborhoods in a $d$-layer DropGNN.

**Theorem 3** *There exists a pair of neighborhoods that cannot be distinguished by* 1- *and* 2-*dropouts.*

We illustrate a simpler example for only 1-dropouts in Figure 3. For a construction that also covers the case of 2-dropouts, the analysis is more technical; we defer this to Appendix C.

We note that even these more difficult examples can be distinguished with our dropout approach, based on their $k$-dropouts for larger $k$ values. However, this requires an even higher number of runs: we need to ensure that we can observe a reliable distribution even for these many-node dropouts.

On the other hand, our dropout approach becomes even more powerful if we combine it e.g. with the extension by port numbers introduced in [28]. Intuitively speaking, port numbers allow an algorithm to determine all paths to the removed node in a 1-dropout, which in turn allows us to reconstruct the entire $d$-hop neighborhood of $u$. As such, in this case, 1-complete dropouts already allow us to distinguish any two neighborhoods.

**Theorem 4** *In the setting of Theorem 1, a DropGNN with port numbers can distinguish any two non-isomorphic $d$-hop neighborhoods.*

Finally, we note that the expressive power of DropGNNs in the 1-complete case is closely related to the *graph reconstruction problem*, which is a major open problem in theoretical computer science since the 1940s [14]. We discuss the differences between the two settings in Appendix C.

**Aggregation with `mean`.** We have seen in Section 3.4 that even with `mean` aggregation, DropGNNs can sometimes distinguish 1-hop neighborhoods (that is, multisets $S_1$ and $S_2$ of features) which look identical to a standard GNN. One can also prove in general that a similar separation is possible in various cases, e.g. whenever the two multisets have the same size.

**Lemma 1** *Let $S_1 \neq S_2$ be two multisets of feature vectors with $|S_1| = |S_2|$. Then $S_1$ and $S_2$ can be distinguished by a DropGNN with `mean` neighborhood aggregation.*

However, in the general case, `mean` aggregation does not allow us to separate any two multisets based on 1-dropouts. In particular, in Appendix C, we also describe an example of multisets $S_1 \cap S_2 = \emptyset$ where the distribution of means obtained from 0- and 1-dropouts is essentially identical in $S_1$ and $S_2$. This implies that if we want to distinguish these multisets $S_1$ and $S_2$, then the best we can hope for is a more complex approach based on multiple-node dropouts.

# 5 Experiments

In all cases we extend the base GNN model to a DropGNN by running the GNN $r$ times in parallel, doing mean aggregation over the resulting $r$ node embedding copies before the graph readout step and then applying the base GNN's graph readout. Additionally, an auxiliary readout head is added to produce predictions based on each individual run. These predictions are used for an auxiliary loss term which comprises $\frac{1}{3}$ of the final loss. Unless stated otherwise, we set the number of runs to $m$ and choose the dropout probability to be $p = \frac{1}{m}$, where $m$ is the mean number of nodes in the graphs in the dataset. This is based on the assumption, that in the datasets we use the GNN will usually have the receptive field which covers the whole graph. We implement random node dropout by, in each run, setting all features of randomly selected nodes to $0$. See Appendix E for more details about the experimental setup and dataset statistics. The code is publicly available[1].

## 5.1 Datasets beyond WL

| | GIN | | +Ports | | +IDs | | +Random feat. | | +Dropout | |
|---|---|---|---|---|---|---|---|---|---|---|
| Dataset | Train | Test | Train | Test | Train | Test | Train | Test | Train | Test |
| LIMITS 1 [11] | 0.50 ±0.00 | 0.50 ±0.00 | 0.50 ±0.00 | 0.50 ±0.00 | 1.00 ±0.00 | 0.59 ±0.19 | 0.66 ±0.19 | 0.66 ±0.22 | 1.00 ±0.00 | **1.00 ±0.00** |
| LIMITS 2 [11] | 0.50 ±0.00 | 0.50 ±0.00 | 0.50 ±0.00 | 0.50 ±0.00 | 1.00 ±0.00 | 0.61 ±0.26 | 0.72 ±0.17 | 0.64 ±0.19 | 1.00 ±0.00 | **1.00 ±0.00** |
| 4-CYCLES [20] | 0.50 ±0.00 | 0.50 ±0.00 | 1.00 ±0.01 | 0.84 ±0.07 | 1.00 ±0.00 | 0.58 ±0.07 | 0.75 ±0.05 | 0.77 ±0.05 | 0.99 ±0.03 | **1.00 ±0.01** |
| LCC [29] | 0.41 ±0.09 | 0.38 ±0.08 | 1.0 ±0.00 | 0.39 ±0.09 | 1.00 ±0.00 | 0.42 ±0.08 | 0.45 ±0.16 | 0.46 ±0.08 | 1.00 ±0.00 | **0.99 ±0.02** |
| TRIANGLES [29] | 0.53 ±0.15 | 0.52 ±0.15 | 1.0 ±0.00 | 0.54 ±0.11 | 1.00 ±0.00 | 0.63 ±0.08 | 0.57 ±0.08 | 0.67 ±0.05 | 0.93 ±0.12 | **0.93 ±0.13** |
| SKIP-CIRCLES [6] | 0.10 ±0.00 | 0.10 ±0.00 | 1.00 ±0.00 | 0.14 ±0.08 | 1.00 ±0.00 | 0.10 ±0.09 | 0.16 ±0.11 | 0.16 ±0.05 | 0.81 ±0.28 | **0.82 ±0.28** |

Table 1: Evaluation of techniques that increase GNN expressiveness on challenging synthetic datasets. We highlight the best test scores in bold. Compared to other augmentation techniques DropGNN (GIN +Dropout) achieves high training accuracy but also generalizes well to the test set.

To see the capabilities of DropGNN in practice we test on existing synthetic datasets, which are known to require expressiveness beyond the WL-test. We use the datasets from Sato et al. [29] that are based on $3-$regular graphs. Nodes have to predict whether they are part of a triangle (TRIANGLES) or have to predict their local clustering coefficient (LCC). We test on the two counterexamples LIMITS 1 (Figure 2a) and LIMITS 2 from Garg et al. [11] where we compare two smaller structures versus one larger structure. We employ the dataset by Loukas [20] to classify graphs on containing a cycle of length 4 (4-CYCLES). We increase the regularity in this dataset by ensuring that each node has a degree of 2. Finally we experiment on circular graphs with skip links (SKIP-CIRCLES) by Chen et al. [6], where the model needs to classify if a given circular graph has skip links of length $\{2, 3, 4, 5, 6, 9, 11, 12, 13, 16\}$.

For comparison, we try several other GNN modifications which increase expressiveness. For control, we run a vanilla GNN on these datasets. We then extend this base GNN with (i) ports [28] (randomly assigned), (ii) node IDs [20] (randomly permuted), and (iii) a random feature from the standard normal distribution [29]. The architecture for all GNNs is a 4-layer GIN with `sum` as aggregation and $\varepsilon = 0$. For DropGNN $r = 50$ runs are performed. For the SKIP-CIRCLES dataset we use a 9-layer GIN instead, as the skip links can form cycles of up to 17 hops.

We train all methods for $1,000$ epochs and then evaluate the accuracy on the training set. We then test on a new graph (with new features). We report training and testing averaged across 10 initializations in Table 1. We can see that DropGNN outperforms the competition.

## 5.2 Sensitivity analysis

We investigate the impact of the number of independent runs on the overall accuracy. Generally, we expect an increasing number of runs to more reliably produce informative dropouts. We train with a sufficiently high number of runs (50) with the same setting as before. Now, we reevaluate DropGNN but limit the runs to a smaller number. We measure the average accuracy over 10 seeds with 10 tests each and plot this average in Figure 4 on three datasets: LIMITS 1 (Figure 4a), 4-CYCLES (Figure 4b), and TRIANGLES (Figure 4c). In all three datasets, more runs directly translate to higher accuracy.

Next, we investigate the impact of the dropout probability $p$. We use the same setting as before, but instead of varying the number of runs in the reevaluation, we train and test with different probabilities

---

[1]`https://github.com/KarolisMart/DropGNN`

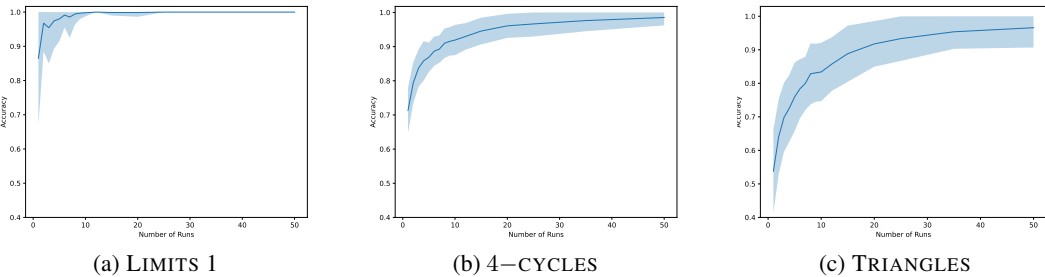

(a) LIMITS 1           (b) 4−CYCLES           (c) TRIANGLES

Figure 4: Investigating the impact of the number of runs ($x$−axis) versus the classification accuracy ($y-axis$). In all three plots, having more runs allows for more stable dropout observations, increasing accuracy. The tradeoff is higher runtime since the model computes more runs.

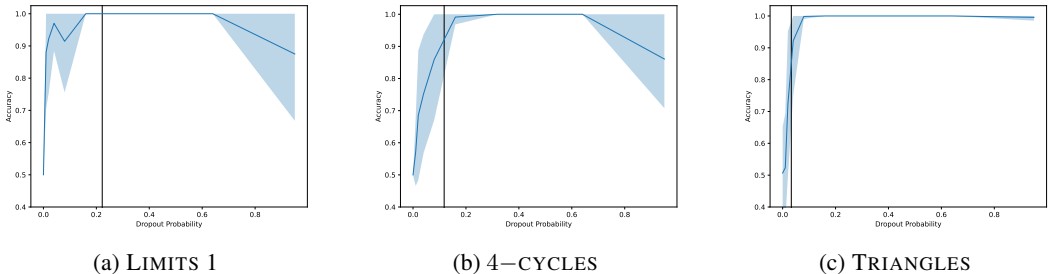

(a) LIMITS 1           (b) 4−CYCLES           (c) TRIANGLES

Figure 5: Investigating the impact of the dropout probability ($x$−axis) versus the classification accuracy ($y$−axis). DropGNN is robust to the choice of $p$ for decently small $p$. Choosing $p \approx \gamma^{-1}$ is a decent default that is shown by vertical black lines.

$p$ on an exponential scale from $0.01$ to $0.64$. We also try $0$ (no dropout) and $0.95$ (almost everything is dropped). Figure 5 shows the accuracy for each dropout probability, again averaged over 10 seeds with 10 tests each. Generally, DropGNN is robust to different values of $p$ until $p$ becomes very large.

## 5.3 Graph classification

| Model | Complexity | MUTAG | PTC | PROTEINS | IMDB-B | IMDB-M |
|---|---|---|---|---|---|---|
| WL subtree [37; 31] | $O(n)$ | 90.4 ±5.7 | 59.9 ±4.3 | 75.0 ±3.1 | 73.8 ±3.9 | 50.9 ±3.8 |
| DCNN [2] | $O(n)$ | - | - | 61.3 ±1.6 | 49.1 ±1.4 | 33.5 ±1.4 |
| PatchySan [23] | $O(n)$ | 89.0 ±4.4 | 62.3 ±5.7 | 75.0 ±2.5 | 71.0 ±2.3 | 45.2 ±2.8 |
| DGCNN [39] | $O(n)$ | 85.8 ±1.7 | 58.6 ±2.5 | 75.5 ±0.9 | 70.0 ±0.9 | 47.8 ±0.9 |
| GIN [37] | $O(n)$ | 89.4 ±5.6 | 64.6 ±7.0 | 76.2 ±2.8 | 75.1 ±5.1 | **52.3 ±2.8** |
| DropGIN (ours) | $O(rn), r \approx 20$ | **90.4 ±7.0** | **66.3 ±8.6** | **76.3 ±6.1** | **75.7 ±4.2** | 51.4 ±2.8 |
| 1-2-3 GNN [22] | $O(n^4)$ | 86.1 | 60.9 | 75.5 | **74.2** | 49.5 |
| PPGN [21]* | $O(n^3)$ | **90.6 ±8.7** | **66.2 ±6.5** | **77.2 ±4.7** | 73 ±5.8 | **50.5 ±3.6** |

Table 2: Graph classification accuracy (%). The best performing model in each complexity class is highlighted in bold. *We report the best result achieved by either of the three versions of their model.

We evaluate and compare our modified GIN model (DropGIN) with the original GIN model and other GNN models of various expressiveness levels on real-world graph classification datasets. We use three bioinformatics datasets (MUTAG, PTC, PROTEINS) and two social networks (IMDB-BINARY and IMDB-MULTI) [38]. Following [37] node degree is used as the sole input feature for the IMDB datasets, while for the bioinformatics datasets the original categorical node features are used.

We follow the evaluation and model selection protocol described in [37] and report the 10-fold cross-validation accuracies [38]. We extend the original 4-layer GIN model described in [37] and use the same hyper-parameter selection as [37]. From Figure 5 we can see that it is usually safer to use a slightly larger $p$ than a slightly smaller one. Due to this, we set the node dropout probability to $p = \frac{2}{m}$, where $m$ is the mean number of nodes in the graphs in the dataset.

Our method successfully improves the results achieved by the original GIN model on the bioinformatics datasets (Table 2) and is, in general, competitive with the more complex and computationally expensive expressive GNNs. Namely, the 1-2-3 GNN [22] which has expressive power close to that of 3-WL and $O(n^4)$ time complexity, and the Provably Powerful Graph Network (PPGN) [21] which has 3-WL expressive power and $O(n^3)$ time complexity. Compared to that, our method has only $O(rn)$ time complexity. However, we do observe, that our approach slightly underperforms the original GIN model on the IMDB-M dataset. Since the other expressive GNNs also underperform when compared to the original GIN model, it is possible that classifying graphs in this dataset rarely requires higher expressiveness. In such cases, our model can lose accuracy compared to the base GNN as many runs are required to achieve a fully stable dropout distribution.

## 5.4 Graph property regression

| Property | Unit | MPNN [34] | 1-GNN [22] | 1-2-3 GNN [22] | PPGN [21] | DropMPNN | Drop-1-GNN |
|---|---|---|---|---|---|---|---|
| $\mu$ | Debye | 0.358 | 0.493 | 0.473 | 0.0934 | **0.059**\* | 0.453\* |
| $\alpha$ | Bohr$^3$ | 0.89 | 0.78 | 0.27 | 0.318 | **0.173**\* | 0.767\* |
| $\epsilon_{HOMO}$ | Hartree | 0.00541 | 0.00321 | 0.00337 | **0.00174** | 0.00193\* | 0.00306\* |
| $\epsilon_{LUMO}$ | Hartree | 0.00623 | 0.00350 | 0.00351 | 0.0021 | **0.00177**\* | 0.00306\* |
| $\Delta\epsilon$ | Hartree | 0.0066 | 0.0049 | 0.0048 | 0.0029 | **0.00282**\* | 0.0046\* |
| $\langle R^2 \rangle$ | Bohr$^2$ | 28.5 | 34.1 | 22.9 | 3.78 | **0.392**\* | 30.83\* |
| ZPVE | Hartree | 0.00216 | 0.00124 | 0.00019 | 0.000399 | **0.000112**\* | 0.000895\* |
| $U_0$ | Hartree | 2.05 | 2.32 | 0.0427 | **0.022** | 0.0409\* | 1.80\* |
| $U$ | Hartree | 2.0 | 2.08 | 0.111 | **0.0504** | 0.0536\* | 1.86\* |
| $H$ | Hartree | 2.02 | 2.23 | 0.0419 | **0.0294** | 0.0481\* | 2.00\* |
| $G$ | Hartree | 2.02 | 1.94 | **0.0469** | 0.24 | 0.0508\* | 2.12 |
| $C_v$ | cal/(mol K) | 0.42 | 0.27 | 0.0944 | **0.0144** | 0.0596\* | 0.259\* |

Table 3: Mean absolute errors on QM9 dataset [25]. Best performing model is in bold and DropGNN versions that improve over the corresponding base GNN are marked with a *.

We investigate how our dropout technique performs using different base GNN models on a different, graph regression, task. We use the QM9 dataset [25], which consists of 134k organic molecules. The task is to predict 12 real-valued physical quantities for each molecule. In all cases, a separate model is trained to predict each quantity. We choose two GNN models to augment: MPNN [12] and 1-GNN [22]. We set the DropGNN run count and node dropout probability the same way as done for graph classification. Following previous work [22; 21] the data is split into $80\%$ training, $10\%$ validation, and $10\%$ test sets. Both DropGNN model versions are trained for 300 epochs.

From Table 3 we can see that Drop-1-GNN improves upon 1-GNN in most of the cases. In some of them, it even outperforms the much more computationally expensive 1-2-3-GNN, which uses higher-order graphs and has three times more parameters [22]. Meanwhile, DropMPNN always substantially improves on MPNN, often outperforming the Provably Powerful Graph Network (PPGN), which as you may recall scales as $O(n^3)$. This highlights the fact that while the DropGNN usually improves upon the base model, the final performance is highly dependent on the base model itself. For example, 1-GNN does not use skip connections, which might make retaining detailed information about the node's extended neighborhood much harder and this information is crucial for our dropout technique.

## 6 Conclusion

We have introduced a theoretically motivated DropGNN framework, which allows us to easily increase the expressive power of existing message passing GNNs, both in theory and practice. DropGNNs are also competitive with more complex GNN architectures which are specially designed to have high expressive power but have high computational complexity. In contrast, our framework allows for an arbitrary trade-off between expressiveness and computational complexity by choosing the number of rounds $r$ accordingly.

**Societal Impact.** In summary, we proposed a model-agnostic architecture improvement for GNNs. We do not strive to solve a particular problem but to enhance the GNN toolbox. Therefore, we do not see an immediate impact on society. We found in our experiments that DropGNN works best on graphs with smaller degrees, such as molecular graphs. Therefore, we imagine that using DropGNN in these scenarios is interesting to explore further.

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
