# Appendix - DropGNN: Random Dropouts Increase the Expressiveness of Graph Neural Networks

**Pál András Papp**
ETH Zurich
apapp@ethz.ch

**Karolis Martinkus**
ETH Zurich
martinkus@ethz.ch

**Lukas Faber**
ETH Zurich
lfaber@ethz.ch

**Roger Wattenhofer**
ETH Zurich
wattenhofer@ethz.ch

## A   Concrete GNN representations for the examples

In this section, we revisit the example graphs from Section 3.4, and we provide a concrete GNN implementation for each of them which is able to distinguish the two cases.

**Example 1.**   Let us assume for simplicity that each node starts with the integer 1 as its single feature. Also, assume that neighborhood aggregation happens with a simple summation, with no non-linearity afterwards, and that this sum is then combined with the node's own feature again through a simple addition.

Now consider this GNN with $d = 2$ layers. Note that in this case, a node $u$ in the left-hand graph is able to gather information from the whole cycle, while a node $u$ in the right-hand graph will behave as if it was the middle node in a simple path of 5 nodes. In both cases, if no dropouts happen, then $u$ will have a value of 3 after the first round, and a value of 9 after the second round.

However, the 1-dropouts are already significantly different: in the left-hand graph, they will produce a result of 5, 5 and 7, while in the right-hand graph, they result in a final value of 5, 5, 8 and 8. One can similarly compute the $k$-dropouts for $k \geq 2$, which will also produce a range of other values (but at most 7 in any case).

If we apply a more sophisticated transformation on these embeddings before run aggregation, then it is straightforward to separate these two distributions. For example, we can use an MLP to only obtain a positive value in case if the embedding is 8 (we discuss this technique in more detail at Example 2); this will happen regularly for the right-hand graph, but never for the left-hand graph. After this, a simple sum run aggregation already distinguishes the cases.

However, if one prefers a simpler transformation, then a choice of $\sigma(x - 8)$ also suffices (with $\sigma$ denoting the Heaviside step function). With this transformation, a run aggregation with sum simply counts the number of cases when the final embedding was a 9. Since the probability of the 0-dropout is different in the two graphs, the expected value of this count will also differ by at least $\Omega(p \cdot r)$ after $r$ runs, which makes them straightforward to distinguish.

**Example 2.**   For an elegant representation of Example 2, the most convenient method is to apply a slightly more complex non-linearity for neighborhood aggregation; this allows a very simple representation for everything else in the GNN.

In particular, let us again assume that each node simply starts with an integer 1 as a feature (i.e. not even aware of its degree initially). Furthermore, assume that neighborhood aggregation happens with a simple sum operator; however, after this, we use a more sophisticated non-linearity $\hat{\sigma}$ which ensures $\hat{\sigma}(2) = 1$, and $\hat{\sigma}(x) = 0$ for all other integers $x$. One can easily implement this function with a 2-layer MLP: we can use $x_1 = \sigma(x - 1)$ and $x_2 = \sigma(-x + 3)$ as two nodes in the first layer, and then combine them with a single node $\sigma(x_1 + x_2 - 1)$ as the second layer. Finally, for the UPDATE function which merges the aggregated neighborhood $x_{N(u)}$ with the node's own embedding $x_u$, let us select $\sigma(x_{N(u)} + x_u - 2)$.

35th Conference on Neural Information Processing Systems (NeurIPS 2021), Sydney, Australia.

The resulting GNN can be described rather easily. Each node begins with a feature of 1, and has an embedding of either 0 or 1 in any subsequent round. The update rule for the embedding is also simple: if $u$'s own value is 1 and $u$ has exactly 2 neighbors with a value of 1, then the embedding of $u$ will remain 1; in any other case, $u$'s embedding is set to 0, and it will remain 0 forever.

In case of dropouts, this GNN will behave differently in the two graphs of Example 2. Note that in both cases, whenever the connected component containing node $u$ is not a cycle after the dropouts, then in at most $d = 3$ rounds, the embedding of $u$ is set to 0. On the other hand, if the component containing $u$ is a cycle, then the embedding of $u$ will remain 1 after any number of rounds.

Now let $u$ denote one of the nodes with degree 3 in both graphs. In the left-hand graph, there is a 1-dropout (of the other gray node) that puts $u$ in a cycle, so $u$ will produce a final embedding of 1 relatively frequently. Besides this, there are also 2 distinct 2-dropouts and a 3-dropout that removes the other gray node but keeps the triangle containing $u$ intact; these will all result in a final embedding of 1 for $u$. On the other hand, in the right-hand graph, there are only 2 distinct 2-dropouts which result in a single cycle containing $u$.

This means that the probability of getting a final value of 1 is significantly higher in the left graph. In particular, after $r$ runs, the difference in the expected frequency of getting a 1 is at least $\Omega(p \cdot r)$, so we can easily separate the two cases by executing run aggregation with `sum` or `mean`.

**Example 3.** The base idea of this separation has already been outlined in Section 3.4: assume that the middle node $u$ uses a simple `mean` aggregation of its neighbors, and the dropout probability is $p = \frac{1}{4}$. Since we are now interested in the behavior of a specific step of `mean` aggregation, we only study the GNN for $d = 1$ rounds.

With $p = \frac{1}{4}$, the left-hand graph provides the following distribution of means in a DropGNN:

$$\Pr(0) = \left(\frac{3}{4}\right)^2 \quad \text{and} \quad \Pr(1) = \Pr(-1) = \frac{1}{4} \cdot \frac{3}{4}.$$

As such, the probability of obtaining a 1 is about $0.19$. Note that we disregarded the case when all neighbors of $u$ are removed, but we could assume for convenience that e.g. the `mean` function also returns 0 in this case. Furthermore, we only considered the cases when $u$ is not removed, since these are the only runs when $u$ computes an embedding at all.

On the other hand, in the right-hand graph, $u$ obtains the following distribution:

$$\Pr(0) = \left(\frac{3}{4}\right)^4 + 4 \cdot \left(\frac{1}{4}\right)^2 \cdot \left(\frac{3}{4}\right)^2 \quad , \quad \Pr\left(\frac{1}{3}\right) = \Pr\left(-\frac{1}{3}\right) = 2 \cdot \frac{1}{4} \cdot \left(\frac{3}{4}\right)^3$$

$$\text{and} \quad \Pr(1) = \Pr(-1) = \left(\frac{1}{4}\right)^2 \cdot \left(\frac{3}{4}\right)^2 + 2 \cdot \left(\frac{1}{4}\right)^3 \cdot \frac{3}{4}.$$

This gives a probability of about $0.06$ for the value 1.

If we apply e.g. the transformation $x \to \sigma(x - 0.5)$ on these values, then the embedding 1 is indeed significantly more frequent in the left-hand graph. Using either `mean` or `sum` for run aggregation allows us to separate the two cases: the final embeddings in the two graphs will converge to $0.19$ and $0.06$ (both multiplied by $r$ in case of `sum`).

**Alternative dropout methods.** Throughout the paper, we consider a natural and straightforward version of the dropout idea: some nodes of the graph (and their incident edges) are removed for an entire run. However, we note that there are also several alternative ways to implement this dropout approach. For example, one could remove edges instead of nodes, or one could remove nodes in an asymmetrical manner (e.g., they still receive, but do not send messages). We point out that all these examples from Section 3.4. could also be distinguished under these alternative models.

# B   Required number of runs

We now discuss the proofs of Theorems 1 and 2.

Note that for any specific subset $S$ of size $k$, the probability of this $k$-dropout happening in a specific run is $p^k \cdot (1-p)^{\gamma+1-k} = \left(\frac{1}{1+\gamma}\right)^k \cdot \left(\frac{\gamma}{1+\gamma}\right)^{\gamma+1-k}$. To obtain the expected frequency $\mathbb{E}X_S$ of this $k$-dropout after $r$ runs, we simply have to multiply this expression by $r$.

Furthermore, given a constant $\delta \in [0, 1]$, a Chernoff bound shows that the probability of significantly deviating from this expected value is

$$\Pr\left( X_S \notin [(1-\delta)\cdot \mathbb{E}X_S, (1+\delta)\cdot \mathbb{E}X_S] \right) \leq 2 \cdot e^{-\frac{\delta^2 \cdot \mathbb{E}X_S}{3}}.$$

Let us consider the case of Theorem 1 first. Since we have $\gamma$ different 1-dropouts, we can use a union bound over these dropouts to upper bound the probability of the event that *any* of the nodes $v \in \Gamma$ will have $X_v \notin [(1-\delta)\cdot \mathbb{E}_1, (1+\delta)\cdot \mathbb{E}_1]$; the probability of this event is at most

$$2 \cdot \gamma \cdot e^{-\frac{\delta^2 \cdot \mathbb{E}_1}{3}}.$$

If we ensure that this probability is at most $\frac{1}{t}$, then the desired property follows. Note that after taking a (natural) logarithm of both sides, this is equivalent to

$$\log(2 \cdot \gamma) - \frac{\delta^2 \cdot \mathbb{E}_1}{3} \leq -\log t,$$

and thus

$$\mathbb{E}_1 \geq \frac{3}{\delta^2} \cdot \log(2 \cdot \gamma \cdot t).$$

Recall that for $\mathbb{E}_1$ we have

$$\mathbb{E}_1 = \frac{1}{1+\gamma} \cdot \left(\frac{\gamma}{1+\gamma}\right)^\gamma \cdot r \geq \frac{1}{1+\gamma} \cdot \frac{1}{e} \cdot r.$$

Due to this lower bound, it is sufficient to ensure

$$\frac{1}{1+\gamma} \cdot \frac{1}{e} \cdot r \geq \frac{3}{\delta^2} \cdot \log(2 \cdot \gamma \cdot t),$$

that is,

$$r \geq \frac{3e}{\delta^2} \cdot (\gamma + 1) \cdot \log(2 \cdot \gamma \cdot t) = \Omega(\gamma \cdot \log \gamma t).$$

This completes the proof of Theorem 1.

For Theorem 2, we also need to upper bound the probability of each dropout combination of multiple nodes. Consider $k$-dropouts for a specific $k$. In this case, we have

$$\mathbb{E}X_S = \left(\frac{1}{1+\gamma}\right)^k \cdot \left(\frac{\gamma}{1+\gamma}\right)^{\gamma+1-k} \cdot r = \frac{1}{\gamma^{k-1}} \cdot \mathbb{E}_1.$$

This implies that in order to ensure $X_S < (1-\delta)\cdot \mathbb{E}_1$ in Theorem 2, it is sufficient to have $X_S < (1-\delta)\cdot \gamma^{k-1} \cdot \mathbb{E}X_S$. If we want to express this as $(1+\epsilon)\cdot \mathbb{E}X_S$ for some $\epsilon$, then we get $\epsilon = (1-\delta)\cdot \gamma^{k-1} - 1$, and thus $\epsilon = \Theta(\gamma^{k-1})$ for appropriately chosen constants. Applying a Chernoff bound (in this case, a different variant that also allows $\epsilon > 1$) then gives

$$\Pr\left( X_S \geq (1+\epsilon)\cdot \mathbb{E}X_S \right) \leq e^{-\frac{\epsilon^2 \cdot \mathbb{E}X_S}{2+\epsilon}}.$$

Since $\epsilon = \Theta(\gamma^{k-1})$ and $\mathbb{E}X_S = \gamma^{-(k-1)} \cdot \mathbb{E}_1$, this is in fact

$$e^{-\Theta(1) \cdot \gamma^{k-1} \cdot \gamma^{-(k-1)} \cdot \mathbb{E}_1} = e^{-\Theta(1) \cdot \mathbb{E}_1}.$$

Note that the number of different $k$-dropouts is $\binom{\gamma}{k} \leq 2^\gamma$, so with a union bound, we can establish this property for each $k$-dropout simultaneously; for this, we need to multiply this error probability by $2^\gamma$. Finally, since we want to ensure this for all $k \geq 2$, we can take a union bound over $k \in \{2, 3, ..., \gamma\}$, getting another multiplier of $\gamma$. Thus to obtain the second condition in Theorem 2 with error probability $\frac{1}{t}$, we need

$$\gamma \cdot 2^\gamma \cdot e^{-\Theta(1) \cdot \mathbb{E}_1} \leq \frac{1}{t}.$$

After taking a logarithm and reorganization, we get

$$\mathbb{E}_1 \ \geq \ \Theta(1) \cdot \log(2^\gamma \cdot \gamma \cdot t)\,.$$

With our lower bound for $\mathbb{E}_1$ and a reorganization of the right side, we can reduce this to

$$r \ \geq \ \Theta(1) \cdot (\gamma + 1) \cdot \gamma \cdot \log(2 \cdot \gamma \cdot t) \ = \ \Omega\left(\gamma^2 + \log \gamma t\right)\,.$$

Another union bound shows that the two conditions of Theorem 2 also hold simultaneously when $r$ is in this magnitude, thus completing the proof of Theorem 2.

Note that if we want to ensure this property for the neighborhood of all the $n$ nodes in the graph simultaneously, then we also have to take a union bound over all the $n$ nodes, which results in a factor of $n$ within the logarithm in our final bounds on $r$.

**Asymptotic analysis.**  Finally, let us note that from a strictly theoretical perspective, if we consider $\gamma$ to be a constant, and $p$ to be some function of $\gamma$, then the probability of any specific $k$-dropout is $p^k \cdot (1 - p)^{\gamma+1-k}$, i.e. a constant value. As such, a Chernoff bound shows that if we select $r$ to be a sufficiently large constant, then every possible dropout combination is observed, and their frequencies are reasonable close to the expected values.

However, this approach is clearly not realistic in practice: e.g. for our choice of $p \approx \gamma^{-1}$, the probability of a specific $k$-dropout is less than $p^k \approx \gamma^{-k}$. This means that we need $r \geq \gamma^k$ runs even to observe this $k$-dropout at least once in expectation. While this $\gamma^k$ is, asymptotically speaking, only a constant value, it still induces a very large overhead in practice, even for relatively small $k$ and $\gamma$ values.

**Different $\gamma$ and $p$ values.**  Note that our choice of $\gamma$ was defined for an arbitrary node of the graph; however, the dropout probability $p$, chosen as a function of $\gamma$, is a global parameter of DropGNNs. As such, our choice of $p$ from the analysis only works well if we assume that the graph is relatively homogeneous, i.e. $\gamma$ is similar for every node.

In practice, one can simply apply the average or the maximum of these different $\gamma$ values; a slightly smaller/larger than optimal $p$ only means that we observe some dropouts with slightly lower probability, or we execute slightly more runs than necessary. The ablation studies in Figures 4 and 5 also show that our approach is generally robust to different number of runs and different dropout probabilities. We note, however, that if e.g. the graph consists of several different but separately homogeneous regions, then a more sophisticated approach could apply a different $p$ value in each of these regions.

## C   Expressiveness with `sum` aggregation

We now discuss our claims on DropGNNs with `sum` neighborhood aggregation. Recall that with this aggregation method, a GNN with injective functions (such as GIN) has the same expressive power as the WL-test.

Note that in this setting, we understand a $d$-hop neighborhood around $u$ to refer to the part of the graph that $u$ can observe in $d$ rounds of message passing. In particular, this contains (i) all nodes that are at most $d$ hops away from $u$, and (ii) all the edges induced by these nodes, except for the edges where both endpoints are exactly at distance $d$ from $u$.

### C.1   Proof of Theorem 3

To prove Theorem 3, we show two different $d$-neighborhoods around a node $u$ (for $d = 2$) that are non-isomorphic, but they generate the exact same distribution of observations for $u$ if we only consider the case of $k$-dropouts for $k \leq 2$.

Note that the example graphs on Figure 3 already provide an example where the 0-dropout and the 1-dropouts are identical. One can easily check this from the figure: in case of no dropouts, $u$ observes the same tree representation in $d = 2$ steps, and in case of any of the 6 possible 1-dropouts (in either of the graphs), $u$ observes the tree structure shown on the right side of the figure.

To also extend this example to the case of 2-dropouts, we need to slightly change it. Note that the example graph is essentially constructed in the following way: we take two independent cycles of

length 3 in one case, and a single cycle of length 6 in the other case, and in both graphs, we connect all these nodes to an extra node $u$. This construction is easy to generalize to larger cycle lengths. In particular, let us consider an integer $\ell \geq 3$, and create the following two graphs: in one of them, we take two independent cycles of length $\ell$, and connect each node to an extra node $u$, while in the other one, we take a single cycle of length $2 \cdot \ell$, and connect each node to an extra node $u$.

We claim that with a choice of $\ell = 5$, this construction suffices for Theorem 3. As before, one can easily verify that $u$ observes the same 2-hop neighborhood in case of no dropouts, and also identical 2-hop neighborhoods for any of the 10 possible 1-dropouts in both graphs. The latter essentially has the same structure as the right-hand tree in Figure 3, except for the fact that the number of degree-3 branches (i.e. the ones on the left side of $u$ in the figure) is now 7 instead of 3.

It only remains to analyze the distribution of 2-dropouts. For this, note that the only information that $u$ can gather in $d = 2$ rounds is the multiset of degrees of its neighbors. In practice, this will depend on the distance of the two removed nodes in the cycles; in particular, we can have the following cases:

1. If the two nodes are neighbors in (one of) the cycle(s), then due to the dropouts, $u$ will have two neighbors of degree 2, and six neighbors of degree 3. There are $2 \cdot \ell = 10$ possible cases to have this dropout combination in both graphs.

2. If the two nodes are at distance 2 in (one of) the cycle(s), then $u$ will have a single neighbor of degree 1, two neighbors of degree 2 and five neighbors of degree 3. This can again happen in $2 \cdot \ell = 10$ different ways in both graphs.

3. If the nodes have distance at least 3 within the same cycle, or they are in different cycles, then the dropout creates four neighbors of degree 2, and four neighbors of degree 3. In the $2 \cdot \ell$ cycle, this can happen in $\frac{2 \cdot \ell \cdot (2 \cdot \ell - 5)}{2} = 2 \cdot \ell^2 - 5 \cdot \ell = 25$ different ways. In case of the two distinct $\ell$-cycles, this cannot happen in a single cycle at all (i.e. for general $\ell$, it can happen in $\frac{\ell \cdot (\ell - 5)}{2}$ ways, but this equals to 0 for $\ell = 5$); however, it can still happen if the two dropouts happen in different cycles, in $\ell \cdot \ell = 25$ different ways.

Hence the distribution of observed neighborhoods is also identical in case of 2-dropouts.

## C.2 Proof of Theorem 4

The setting of Theorem 4 considers GNNs with port numbers (such as CPNGNN) where the neighborhood aggregation function is not permutation invariant, i.e. it can produce a different result for a different ordering of the inputs (neighbors) [12]. Our proof of the theorem already builds on the fact that one can extend the idea of injective GNNs (such as GIN in [15]) to this setting with port numbers. To show that port numbers can be combined with the injective property, one can e.g. apply the same proof approach as in [15], using the fact that the possible combinations of embeddings and port numbers is still a countable set.

Given such an injective GNN with port numbers, the expressiveness of this GNN is once again identical to that of a general distributed algorithm in the message passing model with port numbers [12]. As such, it suffices to show that a distributed algorithm in this model can separate any two different $d$-hop neighborhoods.

Let us assume the 1-complete setting of Theorem 1, i.e. that we have sufficiently many runs to ensure that each 1-dropout is observed at least once in the $d$-hop neighborhood of $u$. We show that the set of neighborhoods observed this way is sufficient to separate any two neighborhoods, regardless of the frequency of multi-dropout cases.

The general idea of the proof is that 1-dropouts are already sufficient to recognize when two nodes in the tree representation of $u$'s neighborhood are actually corresponding to the same node. Consider three nodes $v_1$, $v_2$ and $v_3$, and assume that edges $(v_1, v_3)$ and $(v_2, v_3)$ are both within the $d$-hop neighborhood of $u$. More specifically, assume that $v_1$'s port number $b_1$ leads to $v_3$, and $v_2$'s port number $b_2$ also leads to $v_3$; then we can observe that the nodes at the endpoints of these two edges are always missing from the graph at the same time. That is, since we are guaranteed to observe every 1-dropout at least once, if neighbor $b_1$ of $v_1$ and neighbor $b_2$ of $v_2$ are distinct nodes, then we must observe at least one neighborhood variant where only one of these two neighbors are missing; in this case, we know that the $b_1$th neighbor of $v_1$ and the $b_2$th neighbor of $v_2$ are not identical. On the other

hand, if the two neighbors are always absent simultaneously, then the two edges lead to the same node.

The proof of the theorem happens in an inductive fashion. Note that from the 0-dropout, we can already identify the degree of $u$ in the graph, and the port leading to each of its neighbors; this is exactly the 1-hop neighborhood of $u$.

Now let us assume that we have already reconstructed the $(i-1)$-hop neighborhood of $u$; in this case, we can identify each outgoing edge from this neighborhood by a combination of a *boundary node* (a node at distance $(i-1)$ from $u$) and a port number at this node. We can then extend our graph into the $i$-hop neighborhood of $u$ (for $i \leq d$) with the following two steps:

1. First, we reconstruct the edges going from distance $(i-1)$ nodes to distance $i$ nodes. Let us refer to nodes at distance $i$ as *outer* nodes. Note that all the outer neighbors of the boundary nodes can be identified by the specific outgoing edges from the boundary nodes; we only have to find out which of these outer nodes are actually the same. This can be done with the general idea outlined before: if two boundary nodes $v_1$ and $v_2$ have a neighbor at ports $b_1$ and $b_2$, respectively, and we do not observe a graph variant where only one of these neighbors is missing, then the two edges lead to the same outer node.

2. We also need to reconstruct the adjacencies between the boundary nodes; this is part of the $i$-hop neighborhood of $u$ by definition, but not part of the $(i-1)$-hop neighborhood. This happens with the same general idea as before: assume that $v_2$ and $v_3$ are both nodes at distance $(i-1)$, and $v_1$ is a node at distance $(i-2)$ that is adjacent to $v_3$. Then we can check whether $v_3$ disappears simultaneously from the respective ports $b_1$ and $b_2$ of nodes $v_1$ and $v_2$; if it does, then we know that edge $b_2$ of node $v_2$ leads to this other boundary node $v_3$.

After $d$ steps, this process allows us to reconstruct the entire $d$-hop neighborhood of $u$, thus proving the theorem.

Let us also briefly comment on the GNN interpretation of this graph algorithm. An injective GNN construction ensures that we map different $d$-hop neighborhoods to a different real number embedding. Note that the algorithm can separate any two neighborhoods without using the frequency of the specific neighborhoods variants; this implies that the set of real numbers obtained is different for any two neighborhoods, i.e. there must exist a number $z \in \mathbb{R}$ that is present in one of the distributions, but not in the other. One can then develop an MLP that essentially acts as an indicator for this value $z$, only outputting 1 if the input is $z$; this allows us to separate the two neighborhoods.

Finally, note that our main objective throughout the paper was to compute a different embedding for two different neighborhoods. However, in this setting of Theorem 4, it is also possible to encounter the opposite problem: if two $d$-hop neighborhoods are actually isomorphic, but they have a different assignment of port numbers, then they might produce a different embedding in the end.

We point out that with more sophisticated run aggregation, it is also possible to solve this problem, i.e. to recognize the same neighborhood regardless of the chosen port numbering. In particular, we have seen that in the 1-complete case, the multiset of final embeddings already determines the entire neighborhood around $u$, and thus also its isomorphism class. This means that there is a well-defined function from the embedding vectors in $\mathbb{R}^r$ that we can obtain in $r$ runs to the possible isomorphism classes of $u$'s neighborhood (assuming for convenience that the neighborhood size is bounded). Due to the universal approximation theorem, a sufficiently complex MLP can indeed implement this function; as such, determining the isomorphism class of $u$'s neighborhood is indeed within the expressive capabilities of DropGNNs in this setting. However, while such a solution exists in theory, we note that this graph isomorphism problem is known to be rather challenging in practice.

### C.3 Briefly on the graph reconstruction problem

The graph reconstruction problem is a well-known open question dating back to the 1940s. Assume that there is a hidden graph $G$ on $n \geq 3$ nodes that we are unaware of; instead, what we receive as an input is $n$ different modified variants of $G$, each obtained by removing a different node (and its incident edges) from $G$. This input multiset of graphs is often called the *deck* of $G$. Note that the graphs in the deck are only provided up to an isomorphism class, i.e. for a specific node of the deck graph, we do not know which original node of $G$ it corresponds to. The goal is to identify $G$ from its

deck; this problem is solvable exactly if there are no two non-isomorphic graphs with the same deck. This assumption is known as the graph reconstruction conjecture [5].

This problem is clearly close to our task of reconstructing a neighborhood from its 1-dropout variants; however, there are also two key differences between the settings. Firstly, in our DropGNNs, we do not observe a graph, but rather a tree-representation of its neighborhood where some nodes may appear multiple times. In this sense, our GNN setting is much more challenging than the reconstruction problem, since it is highly non-trivial to decide whether two nodes in this tree representation correspond to the same original node. On the other hand, the DropGNN setting has the advantage that we can also observe the 0-dropout; this does not happen in the reconstruction problem, since it would correspond to directly receiving the solution besides the deck.

## D  Dropouts with `mean` or `max` aggregation

In this section, we discuss the expressiveness of the dropout technique with `mean` and `max` neighborhood aggregation. In particular, we prove that separation is always possible with `mean` aggregation when $|S_1| = |S_2|$, we construct a pair of neighborhoods that provide a very similar distribution of mean values, and we briefly discuss the limits of `max` aggregation in practice.

### D.1  Proof of Lemma 1

We begin with the proof of Lemma 1. More specifically, we show that if $|S_1| = |S_2|$, then there always exists a choice of $p$ and integers $a, b$ such that after applying an activation function $\sigma(ax + b)$ on $S_1$ and $S_2$, a `mean` neighborhood aggregation allows us to distinguish the two sets.

In our proof, we assume that $S_1$ and $S_2$ are both multisets of integers (instead of vectors), i.e. that node features are only 1-dimensional. With multi-dimensional feature vectors, we can apply the same proof to each dimension of the vectors individually; since $S_1 \neq S_2$, we will always have a dimension that allows us to separate the two multisets with the same method.

Let $\overline{s}_1$ denote the mean of $S_1$ and $\overline{s}_2$ denote the mean of $S_2$. We first discuss the simpler case when $\overline{s}_1 \neq \overline{s}_2$; if this holds, we can distinguish any two sets $S_1$ and $S_2$, so we make this proof for the general case, without the assumption that $|S_1| = |S_2|$. After this, we discuss the case when $\overline{s}_1 = \overline{s}_2$ and $|S_1| = |S_2|$; this completes the proof of Lemma 1.

The main idea of the proofs is to find a threshold $\tau$ such that in $S_1$, we have mean values larger than $\tau$ much more frequently than in $S_2$ (or vice versa). We can then use an activation function $\hat{\sigma}(x) := \sigma(x - \tau)$ (with $\sigma$ denoting the Heaviside step function) to ensure that $\sigma(x) = 1$ if $x \geq \tau$, and $\sigma(x) = 0$ otherwise. This means that a run aggregation with `sum` will simply count the cases when the mean is larger than $\tau$, and thus with high probability, we get a significantly different sum in case of $S_1$ and $S_2$.

Note that even though the proof is described with a Heaviside activation function for ease of presentation, one could also use the logistic function (a more popular choice in practice), since the logistic function provides an arbitrary close approximation of the step function with the appropriate parameters.

**When the means are different.**  First we consider the case when $\overline{s}_1 \neq \overline{s}_2$.

In this setting, finding an appropriate $\tau$ is relatively straightforward. Assume w.l.o.g. that $\overline{s}_1 < \overline{s}_2$, and let us choose an arbitrary $\tau$ such that $\overline{s}_1 < \tau < \overline{s}_2$. This implies that whenever no node is removed, then the mean in $S_1$ will produce a 0, while the mean in $S_2$ will produce a 1.

It only remains to ensure that 0-dropouts are frequent enough to distinguish these two cases. For this, let $\gamma = \max(|S_1|, |S_2|)$, and let us select $p = \frac{1}{2\gamma}$. For both $S_1$ and $S_2$, this gives a probability of at least

$$(1 - p)^{\gamma} = \left(\frac{2\gamma - 1}{2\gamma}\right)^{\gamma}$$

for 0-dropouts. When $\gamma \geq 2$, this probability is strictly larger than $0.55$.

With a Chernoff bound, one can also show that the number of 0-dropouts is strictly concentrated around this value: with $\delta = 0.05$ and $r$ runs, the probability of the number of 0-dropouts being below

$(1 - \delta) \cdot 0.55 \approx 0.52$ is upper bounded by $e^{-\frac{1}{3} \cdot \delta^2 \cdot 0.55 \cdot r}$. To ensure that this is below $\frac{1}{t}$, we only need $\Theta(1) \cdot r \geq \log t$, and hence $r \geq \Omega(\log t)$. This already ensures that in case of $S_2$, we have at least $0.52 \cdot r$ runs that produce a 1, while in $S_1$, we have at least $0.52 \cdot r$ runs that produce a 0 (i.e. at most $0.48 \cdot r$ runs that produce a 1). Hence with high probability, a `sum` run aggregation gives a sum below $0.48 \cdot r$ and above $0.52 \cdot r$ for $S_1$ and $S_2$ respectively, so the two cases are indeed separable.

**When the means are the same.**    Now consider the case when $\bar{s}_1 = \bar{s}_2$, and we have $|S_1| = |S_2|$.

In this setting, let $\gamma = |S_1| = |S_2|$. Since the multisets are not identical, there must be an index $i \in \{1, ..., \gamma\}$ such that in the sorted version of the multisets, the $i^{\text{th}}$ element of $S_1$ is different from the $i^{\text{th}}$ element of $S_2$. Let us consider the smallest such index $i$, and assume w.l.o.g. that the $i^{\text{th}}$ element of $S_1$ (let us call it $x_{1,i}$) is larger than the $i^{\text{th}}$ element of $S_2$ (denoted by $x_{2,i}$). Furthermore, Let $\bar{s}_{1,-i}$ and $\bar{s}_{2,-i}$ denote the mean of $S_1$ and $S_2$, respectively, after removing the $i^{\text{th}}$ element.

Note that if we only had 1-dropouts and 0-dropouts in our GNNs, then finding this index $i$ would already allow a separation in a relatively straightforward way. Since $x_{1,i} > x_{2,i}$, we must have $\bar{s}_{1,-i} < \bar{s}_{2,-i}$. The idea is again to select a threshold value $\tau$ such that $\bar{s}_{1,-i} < \tau < \bar{s}_{2,-i}$. This ensures that in $S_1$, at least $i$ of the 1-dropouts produce a 0, whereas in $S_2$, at most $i - 1$ of the 1-dropouts produce a 0. If the frequency of all 1-dropouts is concentrated around its expectation, then this shows that the occurrences of 1 will be significantly higher in $S_2$.

What makes this argument slightly more technical is the presence of $k$-dropouts for $k \geq 2$. In order to reduce the relevance of these cases, we select a smaller $p$ value. In particular, let $p = \frac{1}{2\gamma^2}$. In this case, the probability of a $k$-dropout is only

$$p^k \cdot (1-p)^{\gamma-k} \leq p^k = \frac{1}{2^k \cdot \gamma^{2k}} \, ,$$

and the probability of having any multiple-dropout case in a specific run is at most

$$\sum_{k=2}^{\gamma} \binom{\gamma}{k} \cdot \frac{1}{2^k \cdot \gamma^{2k}} \leq \sum_{k=2}^{\gamma} \frac{\gamma^k}{2} \cdot \frac{1}{2^k \cdot \gamma^{2k}} \leq \sum_{k=2}^{\gamma} \frac{1}{2^{k+1}} \cdot \frac{1}{\gamma^k} \leq \frac{1}{4 \cdot \gamma^2} \, ,$$

using the fact that $\binom{\gamma}{k} \leq \frac{1}{2} \cdot \gamma^k$ for $k \geq 2$ and the fact that $\frac{1}{8} + \frac{1}{16} + ... \leq \frac{1}{4}$.

On the other hand, the probability of a 1-dropout is

$$p \cdot (1-p)^{\gamma-1} = \frac{1}{2\gamma^2} \cdot \left( \frac{2\gamma^2 - 1}{2\gamma^2} \right)^{\gamma-1} \, ,$$

where one can observe that the second factor is at least $\frac{7}{8}$ for any positive integer $\gamma$. As such, the probability of a 1-dropout is lower bounded by $\frac{7}{16} \cdot \frac{1}{\gamma^2}$, i.e. it is notably larger than the cumulative probability of multiple-dropout cases.

This means that our previous choice of $\bar{s}_{1,-i} < \tau < \bar{s}_{2,-i}$ also suffices for this general case. In particular, even if all the multiple-dropouts in $S_1$ produce a mean that is larger than $\tau$, and all the multiple-dropouts in $S_2$ produce a mean that is smaller than $\tau$, we will still end up with a considerably larger probability of obtaining a value of 1 in case of $S_2$, due to the 1-dropout of the $i^{\text{th}}$ element. More specifically, the difference between the two probabilities will be at least $\frac{3}{16} \cdot \frac{1}{\gamma^2}$; using a Chernoff bound in a similar fashion to before, one can conclude that $\Omega(\gamma^4 \cdot \log t)$ runs are already sufficient to separate the two case with error probability at most $\frac{1}{t}$.

### D.2   Construction for similar mean distribution

Let us now comment on the general case when we have $\bar{s}_1 = \bar{s}_2$ but $|S_1| \neq |S_2|$. We present an example for two different sets $S_1$ and $S_2$ where the distribution of mean values obtained from 0- and 1-dropouts is essentially identical, thus showing the limits of any general approach that uses `mean` aggregation, but does not execute a deeper analysis of $k$-dropouts for $k \geq 2$.

Consider an even integer $\ell$, and consider the following two subsets. Let $S_1$ consist of $\frac{\ell}{2}$ distinct copies of the number $-(\ell-1)$, and $\frac{\ell}{2}$ distinct copies of the number $(\ell-1)$. Let $S_2$ consist of $\frac{\ell}{2}$ distinct copies of the number $-\ell$, and $\frac{\ell}{2}$ distinct copies of the number $\ell$, and a single instance of 0. These sets

provide $|S_1| = \ell$ and $|S_2| = \ell + 1$, and also $\bar{s}_1 = \bar{s}_2 = 0$. For a concrete example of $\ell = 4$, we get the multisets $S_1 = \{-3, -3, 3, 3\}$ and $S_2 = \{-4, -4, 0, 4, 4\}$.

The mean values obtained for 1-dropouts is also easy to compute in these examples. In $S_1$, we have $\frac{\ell}{2}$ distinct 1-dropouts with a mean of 1, and $\frac{\ell}{2}$ distinct 1-dropouts with a mean of $-1$. In $S_2$, we have $\frac{\ell}{2}$ distinct 1-dropouts with a mean of 1, and $\frac{\ell}{2}$ distinct 1-dropouts with a mean of $-1$, and a single 1-dropout with a mean of 0.

Note that if we only consider these 0 and 1-dropouts, then the probability of getting a 0 is exactly the same in both settings. In $S_1$, this comes from the probability of the 0-dropout only, so it is $(1-p)^\ell$. In $S_2$, we have to add up the probability of the 0-dropout and a single 1-dropout: this is $(1-p)^{\ell+1} + p \cdot (1-p)^\ell = (1-p)^\ell$.

The set of means obtained from 1-dropouts is also identical in the two neighborhoods, it is only their probability that is slightly different. In $S_1$, both $-1$ and $1$ are obtained with probability $\frac{\ell}{2} \cdot p \cdot (1-p)^{\ell-1}$, while in $S_2$, they are both obtained with probability $\frac{\ell}{2} \cdot p \cdot (1-p)^\ell$. Hence the difference between the two probabilities is only

$$\frac{\ell}{2} \cdot p \cdot \left((1-p)^{\ell-1} - (1-p)^\ell\right) = \frac{\ell}{2} \cdot p^2 \cdot (1-p)^{\ell-1}.$$

Recall that we have $\Theta(\ell^2)$ distinct 2-dropouts, each with a probability of $p^2 \cdot (1-p)^{\ell-1}$, so these 2-dropouts are together easily able to bridge this difference of frequency of the 1-dropouts between $S_1$ and $S_2$. This shows that we cannot conveniently ignore multiple-node dropouts as in case of $|S_1| = |S_2|$ before: the only possible 1-dropout-based approach to separate the two sets (i.e. to use the slightly different frequency of the values $-1$ and $1$) is not viable without a deeper analysis of the distributions of 2-dropouts. It is beyond the scope of this paper to analyze this distribution in detail, or to come up with more sophisticated separation methods based on multiple-node dropouts.

### D.3 Aggregation with `max`

Another well-known permutation-invariant function (and thus a natural candidate for neighborhood aggregation) is `max`; however, this method does not combine well with the dropout approach in practice.

In particular, if the multisets $S_1$ and $S_2$ only differ in their smallest element, then `max` aggregation can only distinguish them from a specific $(\gamma - 1)$-dropout when all other neighbors of $u$ are removed. This dropout combination only has a probability of $p^{\gamma-1} \cdot (1-p)^2$; thus for a reasonably small $p$ (e.g. for $p \approx \gamma^{-1}$), we need a very high number of runs to observe this case with a decent probability.

## E    Details of the experimental setup

In all of our experiments, we use Adam optimizer [8]. For synthetic benchmarks and graph classification, we use a learning rate of $0.01$, for graph property regression we use a learning rate of $0.001$. For graph classification benchmarks we decay the learning rate by 0.5 every 50 steps [15] and for the graph regression benchmark we decay the learning rate by a factor of 0.7 on plateau [10]. The GIN model always uses 2-layer multilayer perceptrons and batch normalization [7] after each level [15]. For our dropout technique, during preliminary experiments we tested three different node dropout implementation options: i) completely removing the dropped nodes and their edges from the graph; ii) replacing dropped node features by 0s before and after each graph convolution; iii) replacing the initial dropped node features by 0s. These preliminary experiments showed that all of these options performed similarly in practice, but the last option resulted in a more stable training. Since it is also the simplest dropout version to implement we chose to use it in all of our experiments. To ensure that the base model is well trained, when our technique is used we apply an auxiliary loss on each run individually. This auxiliary loss comprises $\frac{1}{3}$ of the final loss. While our model can have $O(n)$ memory consumption if we execute the runs in sequence, we implement it in a paralleled manner, which reduces the compute time, as all $r$ runs are performed in parallel, but increases memory consumption.

For the synthetic benchmarks (LIMITS 1, LIMITS 2, 4-CYCLES, LCC, TRIANGLES, SKIP-CIRCLES) we use a GIN model with 4 convolutional layers (+ 1 input layer), `sum` as aggregation, $\varepsilon = 0$ and for

simplicity do not use dropout on the final READOUT layer, while the final layer dropout is treated as a hyper-parameter in the original model. For synthetic node classification tasks (LIMITS 1, LIMITS 2, LCC, and TRIANGLES) we use the same readout head as the original GIN model but skip the graph aggregation step. In all cases, except the SKIPCIRCLES dataset, 16 hidden units are used for synthetic tasks. For the SKIPCIRCLES dataset we use a GIN model with 9 convolutional layers (+ 1 input layer) with 32 hidden units as this dataset has cycles of up to 17 hops and requires long-range information propagation to solve the task. For the DropGIN variant, `mean` aggregation is used to aggregate node representations from different runs. When the GIN model is augmented with ports, which introduce edge features, we use modified GIN convolutions that include edge features [6]. In synthetic benchmarks, we always generate the same number of graphs for training and test sets (generate a new copy of the dataset for testing) and for each random seed, we re-generate the datasets. We always feed in the whole dataset as one batch. LIMITS 1, LIMITS 2 and SKIP-CIRCLES datasets are always comprised of graphs with the same structure, just with permuted node IDs for each dataset initialization, the remaining datasets have random graph structure, which changes when the datasets are regenerated. You can see the synthetic dataset structure type and statistics in Table 5. All nodes in these datasets have the same degree.

| Dataset | Number of graphs | Number of nodes | Degree | Structure | Task |
|---------|------------------|-----------------|--------|-----------|------|
| LIMITS 1 [3] | 2 | 16 | 2 | Fixed | Node classification |
| LIMITS 2 [3] | 2 | 16 | 3 | Fixed | Node classification |
| 4-CYCLES [9] | 50 | 16 | 2 | Random | Graph classification |
| LCC [13] | 6 | 10 | 3 | Random | Node classification |
| TRIANGLES [13] | 1 | 60 | 3 | Random | Node classification |
| SKIP-CIRCLES [1] | 10 | 41 | 4 | Fixed | Graph classification |

Table 5: Synthetic dataset statistics and properties.

For graph classification tasks we use exactly the same GIN model as described originally and apply our dropout technique on top. Namely, with 1 input layer, 4 convolution layers with `sum` as aggregation and $\varepsilon = 0$ and dropout [14] on the final READOUT layer. For the DropGIN variant, `mean` aggregation is used to pool node representations from different runs. Note, that in our setting `sum` and `mean` aggregations are equivalent, up to a constant multiplicative factor, as the number of runs is a constant chosen on a per dataset level. We use exactly the same model training and selection procedure as described by [15]. We decay the learning rate by 0.5 every 50 epochs and tune the number of hidden units $\in \{16, 32\}$ for bioinformatics datasets while using $64$ for the social graphs. The dropout ratio $\in \{0, 0.5\}$ after the final dense layer the batch size $\in \{32, 128\}$ are also tuned. The epoch with the best cross-validation accuracy over the 10 folds is selected. You can see the statistics of synthetic datasets in Table 6.

| Dataset | Number of graphs | Number of nodes | | | Degree | | |
|---------|------------------|-----|-----|------|-----|-----|------|
| | | Min | Max | Mean | Min | Max | Mean |
| MUTAG | 188 | 10 | 28 | 18 | 3 | 4 | 3.01 |
| PTC | 344 | 2 | 64 | 14 | 1 | 4 | 3.18 |
| PROTEINS | 1109 | 4 | 336 | 38 | 3 | 12 | 5.78 |
| IMDB-B | 996 | 12 | 69 | 19 | 11 | 68 | 18.49 |
| IMDB-M | 1498 | 7 | 63 | 13 | 6 | 62 | 11.91 |
| QM9 | 130 831 | 3 | 29 | 18 | 2 | 5 | 3.97 |

Table 6: Real-world dataset statistics.

For the graph property regression task (QM9) we augment two models: 1-GNN [10] and MPNN [4]. For 1-GNN we use the code and the training setup as provided by the original authors[1]. For MPNN we use the reference model implementation from PyTorch Geometric [2]. We otherwise follow the training and evaluation procedure used by 1-GNN [10]. The models are trained for 300 epochs and the epoch with the best validation score is chosen.

We use PyTorch [11] and PyTorch Geometric [2] for the implementation. All models have been trained on Nvidia Titan RTX GPU (24GB RAM).

---

[1] https://github.com/chrsmrrs/k-gnn
[2] https://github.com/rusty1s/pytorch_geometric/blob/master/examples/qm9_nn_conv.py