# OpenReview forum: "DropGNN: Random Dropouts Increase the Expressiveness of Graph Neural Networks"
_NeurIPS.cc/2021/Conference — NeurIPS 2021 Spotlight_

### Official Review · Reviewer_71WX · 2021-07-09

**Rating:** 7
**Confidence:** 4

**Summary:**

The paper suggests a method to the GNN toolbox. The proposed method applies multiple, independent evaluations of a graph, each including random dropout applied to the graph nodes. The different evaluations are then aggregated to a single classification/regression result.

**Limitations And Societal Impact:**

As mentioned above, I would like to see the scalability of the method addressed.
I have no concerns regarding potential negative societal impact.

**Main Review:**

I find this paper's contributions to be clear. The main idea and ways of implementations are easy to understand, and the empirical results seem adequate.
A few points regarding the paper:
1. (line 135) Does the "simple transformation on each node" has any contribution to the method?
2. I'm not confident this method can scale to large graphs. The authors refer to $\gamma$ almost as a constant. One of the upside of the method is that its runtime efficiency is low compared to other, stronger than 2-WL methods, though tackling larger graphs (not feasible for other methods) is likely to result in a very large $\gamma$.  As a result, to theoretical $r$ needed is also very large.
3. Figure 5 mentions choosing $p=\gamma^{-1}$, but $\gamma$ is defined by an arbitrary graph node. Which $\gamma$ should be used?
4. I'd like to see the some information of the number of graph nodes in each of the dataset used, both in table 1 and 2, to understand how well the method preforms on different sizes of graphs.
5. A runtime comparison of the method and a baseline from $O(n)$, $O(n^{3/4})$, or the method with itself for different $r$ will help understand the computational vs accuracy tradeoff.

**Time Spent Reviewing:**

4

---

> ### Author Response · Authors · 2021-08-10
> **Response to Reviewer 71WX**
>
> Thank you for your review and the thoughtful comments. You can find the answers below and in our general response.
>
> > (line 135) Does the "simple transformation on each node" has any contribution to the method?
>
> As briefly noted in lines 130-133, simply summing up the embeddings from each run is often not expressive enough, e.g. if $d$ is small. As such, we allow DropGNN to first execute a learnable transformation on these embeddings before taking a sum over the different runs. Our examples show that a simple nonlinear transformation step ($x \rightarrow \sigma(Wx+b)$) is already enough to obtain expressive DropGNNs; however, one can also make this transformation step more complex, e.g. a multi-layer perceptron.
>
> > I'm not confident this method can scale to large graphs. The authors refer to $\gamma$ almost as a constant. One of the upsides of the method is that its runtime efficiency is low compared to other, stronger than 2-WL methods, though tackling larger graphs (not feasible for other methods) is likely to result in a very large $\gamma$. As a result, the theoretical $r$ needed is also very large.
>
> We answer this question in the general response in more detail. For graphs with bounded degree, the neighborhood size is independent of $n$ and so is $r$. In such cases, DropGNN scales linearly as simpler GNNs (modulo $r$ as a constant factor). While many graphs do not have bounded degrees, graphs in biology or chemistry often do.
>
> > Figure 5 mentions choosing $p=\gamma^{-1}$ , but $\gamma$ is defined by an arbitrary graph node. Which $\gamma$ should be used?
>
> That is indeed an interesting question. For a simple and robust dropout approach, we assumed that the graph is relatively homogeneous, so the optimal choice of $\gamma$ (and hence $p$) is similar for every node. In practice, one can instead apply the average or the maximum of these different $\gamma$-s. Recall that a slightly smaller/larger than optimal $\gamma$ only means that we observe some dropouts with slightly lower probability, or we execute slightly more runs than necessary.
> The ablation studies in Figures 4 and 5 show that the method is generally robust to the different number of runs and the different dropout probabilities.
> We note, however, that if e.g. the graph consists of several different but separately homogenous regions, then a more sophisticated approach could also apply a different $p$ value in each of these regions.
>
>
> > I'd like to see the some information of the number of graph nodes in each of the dataset used, both in table 1 and 2, to understand how well the method preforms on different sizes of graphs.
>
> ### Synthetic Data (Table 1)
> | Dataset     | Number of nodes |
> |-------------|-----------------|
> | Limits 1    | 16              |
> | Limits 2    | 16              |
> | 4-cycles    | 25              |
> | LCC         | 10              |
> | Triangles   | 60              |
> | Skipcircles | 41           |
>
> ### Real-World Data (Table 2)
> |Dataset   | Mean number of nodes  | Minimum number of nodes  | Maximum number of nodes  |
> |---|---|---|---|
> | MUTAG  | 18  | 10 | 28  |
> | PTC  |  14 | 2 | 64 |
> | PROTEINS  | 38  | 4 | 336 |
> | IMDB-B  | 19  | 12 | 69 |
> | IMDB-M  | 13  | 7 | 63 |
>
> QM9 graphs (Appendix F, Table 3) have 18 nodes on average, the smallest graph has 3 nodes and the largest graph has 29 nodes.
>
> > A runtime comparison of the method and a baseline from $O(n)$,  $O(n^{3/4})$, or the method with itself for different $r$ will help understand the computational vs accuracy tradeoff.
>
> The runtime of the method scales as $O(rn)$ for graphs with bounded degree. In practice, we indeed observe linear runtime scaling with $r$. As you can see from Figure 4, there is a point, where increasing the number of runs brings diminishing returns, so a sweet spot that achieves the best computational vs accuracy tradeoff can be found. You can also find more details on scaling in the general response.

---

> > ### Comment · Reviewer_71WX · 2021-08-24
> > **Response to the rebuttal**
> >
> > Thank you for your response. My concerns were addressed, my rating remains unchanged.

---

### Official Review · Reviewer_LU6D · 2021-07-13

**Rating:** 7
**Confidence:** 3

**Summary:**

The authors propose DropGNN, a GNN exension where multiple forward passes of the model are aggregated. In each of the passes, each node in the graph is dropped with probability $p$ independently. Aggregating a sufficiently large number of dropout combinations effectively breaks symmetries that lead to traditional GNNs to not be able to distinguish between certain neighborhoods. The authors present theoretical results that DropGNNs can distinguish neighborhoods that traditional GNNs (e.g., GIN) cannot. The authors evaluate their method on both synthetic and real-world datasets.

**Limitations And Societal Impact:**

The authors do not mention limitations in the main text. They do have a section in the Appendix, however this is not completely fair to authors who actually used scarce main-text space to highlight their limitations.

**Main Review:**

Strengths:
* The proposed method is conceptually simple yet comes with theoretical advantages over traditional GNNs.
* The method is relatively efficient compared to extensions such as 1-2-3-GNNs.
* Experiments on synthetic datasets suggest that the method can improve upon the baselines.

Weaknesses:
* The evaluation on real-world datasets is not completely convincing; the improvements over the baseline are typically very small compared to the confidence intervals denoted by ±.
* There is no study of the runtime of the approach.
* In general, the paper mentions and explores relatively few real-world applications for the approach. While this is not a major problem for a more theoretical paper, it is generally desirable to show real-world impact.

More details:
* Figure 3 is not clear to me. What exactly does the tree structure on the right depict?
* What do the shaded regions in Figures 4 and 5 indicate?
* What are the results when not modifying the 4-CYCLES dataset?
* The approach seems to resemble some similarity to randomized smoothing [Cohen et al. 2019], which has been extended to graphs by [Bojchevski et al. 2020]. Essentially, the expectation of a classifier under some noise distribution is approximated via Monte Carlo sampling, similar to Theorem 1. Can DropGNN be phrased as a randomized smoothing approach?
* What about dropping edges instead of nodes? Effectively we can view node dropping as a special case of dropping edges (i.e., drop all incoming and outgoing edges of a node).
* line 177: typo. transforms => transform

**Time Spent Reviewing:**

5

---

> ### Author Response · Authors · 2021-08-10
> **Response to Reviewer LU6D**
>
> Thank you for your time reviewing our paper and your suggestions. We hope that the following comments below and in the general response help with some clarifications:
>
> > The evaluation on real-world datasets is not completely convincing; the improvements over the baseline are typically very small compared to the confidence intervals denoted by ±.
>
> Those real-world graphs might be graphs where the graph structure does not give more information than aggregating neighbors, in this case, the additional runtime of DropGNN provides very small improvements. On the other hand, Table 1 or Appendix F show examples, where the richer structures are needed and DropGNN achieves a very significant improvement over the baselines. Please refer to the general response for more details.
>
> > There is no study of the runtime of the approach.
>
> Table 2 briefly discusses the asymptotic runtime of the examined GNN variants. DropGNNs offer a reasonable compromise between running time and accuracy, achieving better results than normal GNNs that have linear complexity, and being much faster than expressive GNNs with e.g. cubic runtime such as PPGNNs. In fact, when graphs are degree-bounded (which is often the case in biological or chemical datasets), DropGNNs scale linearly just like standard GNNs (but with a higher constant factor). Please refer to the general answer for a more detailed explanation.
>
> > Figure 3 is not clear to me. What exactly does the tree structure on the right depict?
>
> In $d$ rounds of message passing, a node $v$ in a GNN essentially observes a tree representation of its own $d$-hop neighborhood, where multiple nodes of this tree might correspond to the same node of the graph. For example, Figure 1 in the work of Sato et al. [21] provides a very nice illustration of this tree representation.
> The right side of Figure 3 shows the tree representation observed by $u$ with $d=2$ layers, after a $1$-dropout. Since this representation is identical for any possible $1$-dropout in either of the two left-hand graphs, the two cases cannot be distinguished from $1$-dropouts.
>
> > What do the shaded regions in Figures 4 and 5 indicate?
>
> We average ablation results over 10 runs. The solid lines show the mean accuracy over these 10 runs and the shaded region the mean plus-minus one standard deviation.
>
> > What are the results when not modifying the 4-CYCLES dataset?
>
> The unmodified version of the 4-CYCLES dataset potentially has different degrees for every node. Even vanilla GNNs can fit this information on the training data (similar to how it would use random features). Therefore we modified the dataset to make it more challenging, with nodes that are less distinguishable. Now we can evaluate which methods work in principle (on the training set) and which methods generalize on top (to the test set).
>
> > The approach seems to resemble some similarity to randomized smoothing [Cohen et al. 2019], which has been extended to graphs by [Bojchevski et al. 2020]. Essentially, the expectation of a classifier under some noise distribution is approximated via Monte Carlo sampling, similar to Theorem 1. Can DropGNN be phrased as a randomized smoothing approach?
>
> Indeed, there is definitely a similarity between the two works (thank you for the suggestion, we will add those papers to the discussion of related work).
> Both settings conduct multiple runs with perturbed graph variants and then aggregate the obtained embeddings into a single value in the end. However, the goals are significantly different. In Bojchevski et al's work, the different embeddings are combined in a "smoothing" operation (such as majority voting), which specifically aims to get rid of the atypical/outlying perturbed variants, in order to make the classifier more robust to attacks. On the other hand, in DropGNNs, the main idea is instead to find and identify those perturbed edge cases that are significantly different from the original neighborhood, since these can allow us to distinguish neighborhoods that otherwise seem identical.
> Figure 2b) is a nice example of this. In most cases, we will drop a white node, which does not help to distinguish the two graphs unlike dropping a grey node. The effect could be even more extreme with more white nodes. Yet DropGNN is only concerned about the (rare) case when one grey node is dropped. On the other hand, randomized smoothing works with the averaged/smoothed version of the graph which leans towards white nodes missing.
>
> > What about dropping edges instead of nodes? Effectively we can view node dropping as a special case of dropping edges (i.e., drop all incoming and outgoing edges of a node).
>
> Indeed, there are several alternative ways to implement the dropout idea. For example, one can drop edges instead of nodes, or one can drop nodes in an asymmetrical manner (e.g., they still receive, but do not send messages). The graphs in Section 3.4., for example, can also be easily distinguished under these alternative models. We have decided to focus on our current dropout approach (dropping nodes) because it seemed like the most natural and straightforward implementation of the idea.

---

> > ### Comment · Reviewer_LU6D · 2021-08-21
> > **Reviewer response**
> >
> > Thank you for your thoughtful and detailed response. My concerns were addressed, hence I've increased my score accordingly.

---

### Official Review · Reviewer_YyQ1 · 2021-07-14

**Rating:** 7
**Confidence:** 4

**Summary:**

This paper proposes DropGNN, a novel technique for improving the expressive power of graph neural networks. DropGNN is simple. It removes nodes randomly (even in test time unlike dropout) and aggregates the results of several runs. This paper provides some examples and theoretical results that show when DropGNN improves its expressive power compared to vanilla GNN.

**Limitations And Societal Impact:**

This paper provides some negative theoretical results as well (Theorem 3 and Section D.2.) I couldn't find any direct societal impacts.

**Main Review:**

# Strengthes

* The proposed approach is simple and easy to implement yet has preferable theoretical guarantees.
* DropGNN can control a trade-off between complexity and accuracy via the number $r$ of runs.
* The experiments with synthetic data clearly show the superiority of the proposed method to existing approaches.
* Section 3.4 provides interesting examples. They helped me understand the rationale behind DropGNN.

# Weaknesses

* The improvements on real datasets are marginal (though they are enough for a theoretical paper.)
* While Section 3.4 shows there do exist some graphs that 1-complete dropouts distinguish, it would be better if some theoretical characterizations on what kind of graphs 1-complete dropouts distinguish could be shown.
* DropGNN resembles the existing methods such as rGIN [21] and DropEdge [18]. I would appreciate it if the authors could add more discussions on the differences between them.

# Comments

* The mean and sum aggregations are essentially the same when the sizes of sets are the same (by appropriate scaling). Theorem 5 assumes $|S_1| = |S_2|$. Therefore, the mean aggregation (without dropouts) is as powerful as the sum aggregation (without dropouts) in this case. (Indeed, with dropouts, $|\hat{S}_1| \neq |\hat{S}_2|$ can happen when different numbers of nodes are removed, and the probabilistic discussion is interesting, but) I'm not sure this discussion is a good characterization of the mean aggregation. Example 3 in Section 3.4 shows the mean aggregation works with DropGNN even when $|S_1| \neq |S_2|$. It would be nice if some positive theoretical results of the mean aggregation when $|S_1| \neq |S_2|$ could be shown.

---

## After Discussions

I've read the authors' responses and other reviews. They addressed most of my concerns. I've increased my score from 6 to 7. I vote for acceptance.

**Time Spent Reviewing:**

4

---

> ### Author Response · Authors · 2021-08-10
> **Response to Reviewer YyQ1**
>
> Many thanks for your ideas and comments. Please find our thoughts about them below and in the general response.
> > The improvements on real datasets are marginal (though they are enough for a theoretical paper.)
>
> There are graphs where the graph structure does not give more information than simply aggregating the neighbors, in this case, the additional runtime of DropGNN provides marginal benefits. On the other hand, there exist examples, where the richer structures are needed and DropGNN achieves a very significant improvement over the baselines, which indeed justifies the larger running time of the approach (such as in Table 1 or Appendix F).
>
> > While Section 3.4 shows there do exist some graphs that 1-complete dropouts distinguish, it would be better if some theoretical characterizations on what kind of graphs 1-complete dropouts distinguish could be shown.
>
> Indeed, the characterization of such graphs is a very interesting question. While it is probably out of the scope of the current paper, it sounds like a promising direction for future (theoretical) work on the topic. We are grateful for the suggestion.
>
> > DropGNN resembles the existing methods such as rGIN [21] and DropEdge [18]. I would appreciate it if the authors could add more discussions on the differences between them.
>
> We tried to briefly summarize the main difference of our approach to both rGINs (lines 59-61, caption of Table 1) and DropEdge (lines 70-72); however, we have kept this rather short due to strict space constraints. We will add a more detailed discussion of these differences in the final version of the paper. Let us also elaborate on the difference here:
> * rGINs and DropGNNs have the shared goal of being more expressive than 1WL GNNs. Both models use a source of randomness. While DropGNN does multiple runs with dropping nodes, rGIN augments each node once with a random feature. The problem with this approach is that rGINs learn to calibrate on the specific random values, and as such cannot be used on unseen test graphs that received different random values. On the other hand, DropGNN generalizes to unseen test graphs. One can alleviate this issue of rGIN by training on huge datasets for a long time so that most of the combinatorically many random feature combinations are observed. This makes DropGNN exponentially more data efficient.
> * DropEdge is inspired by Dropout and works as a regularizer to make training more robust. As such, DropEdge does not drop edges in testing. On the other hand, DropGNN drops nodes (and thus edges) to allow distinguishing richer structures. For this, it is important that DropGNN drops nodes during both training and testing.
>
> > The mean and sum aggregations are essentially the same when the sizes of sets are the same (by appropriate scaling). Theorem 5 assumes $S_1 = S_2$. Therefore, the mean aggregation (without dropouts) is as powerful as the sum aggregation (without dropouts) in this case. (Indeed, with dropouts, $S_1 \neq S_2$ can happen when different numbers of nodes are removed, and the probabilistic discussion is interesting, but) I'm not sure this discussion is a good characterization of the mean aggregation. Example 3 in Section 3.4 shows the mean aggregation works with DropGNN even when $S_1 \neq S_2$. It would be nice if some positive theoretical results of the mean aggregation when $S_1 \neq S_2$ could be shown.
>
> We agree that a more general version of Theorem 5 (for any sets $S_1$ and $S_2$) would be a stronger result; however, our negative results (in lines 272-275, and Appendix D.2) show that the generalization of the theorem to this case would be highly non-trivial. We studied different variants of Theorem 5 before writing the paper, and this was the strongest result we could prove. In a camera-ready version, we would consider demoting Theorem 5 to a lemma and putting more focus on the negative result (lines 272-275) instead.

---

### Official Review · Reviewer_6SSD · 2021-07-16

**Rating:** 7
**Confidence:** 3

**Summary:**

The paper proposes Dropout Graph Neural Networks (DropGNNs), an approach that utlizes dropout in a way that allows GNNs to be more expressive. In DropGNN, multiple runs
are conducted where, in each run, the nodes of the graph are dropped out with a certain small probability and passed through a
GNN to produce a distinct embedding. The embeddings of the multiple runs are aggregated (with a sum) after a non-linear
transformation of the node features. DropGNNs achieve competitive results on synthetic and real-world datasets.



**Limitations And Societal Impact:**

The authors have adequately addressed the limitations and impact of their work.

**Main Review:**

### Strengths:

- The idea is simple but clean and interesting; it's also well supported by the theoretical analysis.
- Well-written and accessible paper. Nice examples help illustrate the main point.
- Good generalization results on the synthetic datasets.
- The limitations of DropGNNs are clearly laid out by the authors.
- Nice studies on the number of runs and the dropout probability.
- Additional results in regression experiments.


### Criticism, Comments, and Questions:

- One issue that is not entirely clear to me from the paper is the scaling behavior of this approach.
Given that in both tasks the graphs are relatively small, I think it would be important to see
a study on how this strategy performs and how many runs are needed as the size of the graphs increases, either on synthetic
or on real-world data.

- Modern graph classification benchmarking usually involves some large-scale datasets like the ones in OGB[1] (see also
the models in the leaderboards).
I think it would improve the legitimacy of the results if the authors brought the graph classification comparisons more in line
with those standards.

- Computational cost appears to be an important consideration. DropGNNs will take more time to execute due to the need for multiple runs.
Given that the benefits in graph classification are marginal even in the reported datasets, the tradeoff does not appear particularly enticing unless the size of the instances is small.

- Having a paragraph where the full setup from input to output is described (dimensions of inputs/outputs, aggregation functions, etc.) could help improve clarity.

### Recap:

The paper is well written and accessible, with clearly established limitations and solid theoretical analysis. While the results
on synthetic data show impressive generalization, I am not entirely convinced by the efficacy of this approach when considering the rest of the results on real world graphs and regression. Furthermore, it is also not clear how well the proposed approach will scale,
both in terms of computational cost and accuracy.

Still, I think this is an overall interesting paper, so my current grade is 6, which I am willing to adjust after the rebuttal.


[1]  Hu, Weihua, et al. "Open graph benchmark: Datasets for machine learning on graphs." arXiv preprint arXiv:2005.00687 (2020).


**Time Spent Reviewing:**

10

---

> ### Author Response · Authors · 2021-08-10
> **Response to Reviewer 6SSD**
>
> Thank you for your time and helpful comments. We will try to answer them below and in our general response.
> > One issue that is not entirely clear to me from the paper is the scaling behavior of this approach. Given that in both tasks the graphs are relatively small, I think it would be important to see a study on how this strategy performs and how many runs are needed as the size of the graphs increases, either on synthetic or on real-world data.
>
> > Modern graph classification benchmarking usually involves some large-scale datasets like the ones in OGB[1] (see also the models in the leaderboards). I think it would improve the legitimacy of the results if the authors brought the graph classification comparisons more in line with those standards.
>
> The scaling behavior of DropGNN is mainly determined whether large graphs with more nodes lead to larger neighborhoods. For graphs with bounded degrees, this is not the case. For these graphs, DropGNN scales almost as well as normal GNNs (except it does a constant factor more runs). We explain this in more detail in our general response and provide an additional scaling experiment for illustration. We are also looking into experiments on OGB.
>
> > Computational cost appears to be an important consideration. DropGNNs will take more time to execute due to the need for multiple runs. Given that the benefits in graph classification are marginal even in the reported datasets, the tradeoff does not appear particularly enticing unless the size of the instances is small.
>
> We provide a longer explanation in our general answer. If graph structure is not too important in a dataset, but it is instead enough to e.g. aggregate information from a node's neighbors, DropGNN yields marginal improvements. On the other hand, Table 1 and Appendix F show examples where learning the graph structure is very important and where the dropout approach achieves a very significant improvement over the baselines, which indeed justifies the larger running time of the approach.
>
> > Having a paragraph where the full setup from input to output is described (dimensions of inputs/outputs, aggregation functions, etc.) could help improve clarity.
>
> For the synthetic tests, all nodes have one input feature. For vanilla GIN, GIN with ports, and DropGIN the input feature for each node is set to 1. For GIN with IDs or Random features that node feature is initialized with the corresponding strategy. For real-world datasets, we use the node features provided in the dataset, no edge features are used. For DropGIN, we copy the graph $r$ times, select random nodes in each copy and set their input features to 0, and then run the same GIN model on all of the modified graph copies. Then, we aggregate the final node embeddings over all of the runs. In graph classification tasks the node embeddings are aggregated again using sum pooling to get the final graph embedding. In either case, outputs are then passed through a one-layer MLP to get the final prediction. GIN convolution in all of the experiments uses sum aggregation. The $r$ Dropout runs are aggregated using sum in the synthetic benchmarks, which allows us to do the first ablation study, while for the real world datasets mean aggregation is used, which increases training stability. Note that mean aggregation is equivalent to sum aggregation if we always use the same number of runs. The number of hidden units used, other hyperparameters, and details of the experimental setup are described in Appendix E. We will extend this appendix in the final version of the paper to describe the setup in more detail.

---

> > ### Comment · Reviewer_6SSD · 2021-08-24
> > **Update**
> >
> > I think it could be useful to incorporate parts of the author responses regarding the experimental results either in the main text or in the supplement, since they can help the reader understand in what tasks this method could be expected to perform well.
> >
> > Overall, this is a good submission so I have updated my score accordingly.

---

### Author Response · Authors · 2021-08-10
**General Remarks: Scalability and Experimental Results**

We want to thank all reviewers for their time and input on our paper. We saw two themes that arise in almost all reviews which we want to address jointly:

> What is the scalability of DropGNN / Can we use DropGNN for large graphs?

We provide some asymptotic analysis of DropGNN in Table 2. Compared to normal GNN variants the overhead of DropGNN comes from the factor $r$. The value $r$ is dependent on the neighborhood size. For graphs with bounded degree, $r$ is independent of $n$ and thus the number of necessary runs does not increase with the size of the graph. As such, the dropout method scales very well in this case.

This bounded degree assumption is indeed realistic in many (e.g. biological and chemical) applications and as such, it is also common in other theoretical works on the power of GNNs, e.g. in Sato et al's paper [20]. The synthetic datasets have mostly bounded degrees, too. This suggests that the ablation study in Figure 4 and the derived number of necessary runs should also work identically for graphs of much larger size.

To test this assumption we created a new experiment that trains on the triangles dataset. This dataset consists of 3-regular graphs. For testing, we use graphs of increasing size. We compare the runtime of DropGNN with a GIN and quadratic complexity. We find that DropGNN scales linearly like GIN, albeit with a higher constant factor. We plan to include this experiment and plots of this behavior in the revised version of the paper.

>The experimental results do show minor improvements from DropGNN to other GNN methods, which might not be enough to justify the computational cost:

When considering the general datasets in Table 2, the improvements are indeed often small. We hypothesize that in these datasets, the graph structure might not be very important, and it is instead enough to e.g. aggregate information from a node's neighbors; in these cases, the extra running time invested by DropGNN would even be a waste of resources.

However, in more specific applications where the graph structure is a crucial factor, DropGNNs often yield a much more significant improvement. We can see this clearly in Table 1. We have also included a concrete such application (molecule property regression) in Appendix F; one can see that here the dropout approach significantly outperforms the baselines. We see that the results in Table 2 show only marginal improvements; for a final version, we will try to fit some of the molecule regression results within the main part of the paper. Based on your recommendations we also started looking into the OGB datasets.

---

### Decision · Program_Chairs · 2021-09-27

**Decision:**

Accept (Spotlight)

**Comment:**

The paper proposes a simple idea for improving the expressive power of standard graph neural networks at the expense of speed/memory.

* The reviewers agreed that the paper is well-written and that the proposed idea is elegant and simple to implement.

* The evaluation also shows that DropGNN can improve upon baselines relying upon feature augmentation.

Overall, this is a very good work that meaningfully contributes to the growing GNN literature.